# From Binary to Continuous: Stochastic Re-Weighting for Robust Graph Explanation

## Abstract

Graph Neural Networks (GNNs) have achieved remarkable performance in a wide range of graph-related learning tasks. However, explaining their predictions remains a challenging problem, especially due to the mismatch between the graphs used during training and those encountered during explanation. Most existing methods optimize soft edge masks on weighted graphs to highlight important substructures, but these graphs differ from the unweighted graphs on which GNNs are trained. This distributional shift leads to unreliable gradients and degraded explanation quality, especially when generating small, sparse subgraphs. To address this issue, we propose a novel iterative explanation framework which improves explanation robustness by aligning the model's training data distribution with the weighted graph distribution appeared during explanation. Our method alternates between two phases: explanation subgraph identification and model adaptation. It begins with a relatively large explanation subgraph where soft mask optimization is reliable. Based on this subgraph, we assign importance-aware edge weights to explanatory and non-explanatory edges, and retrain the GNN on these weighted graphs. This process is repeated with progressively smaller subgraphs, forming an iterative refinement procedure. We evaluate our method on multiple benchmark datasets using different GNN backbones and explanation methods. Experimental results show that our method consistently improves explanation quality and can be flexibly integrated with different architectures.

## 1 Introduction

Graph Neural Networks (GNNs) have become a powerful tool for learning from graph-structured data, with applications in molecular property prediction (Wu et al., 2018), social networks (Bu & Shin, 2023), and knowledge graphs (Wang et al., 2019). However, they often behave as black-box models, posing challenges for explainability in high-stakes domains such as healthcare (Anklin et al., 2021) and finance (Rai, 2020), where transparency is essential.

To address these concerns, many post-hoc explanation methods have recently been developed to interpret the decision-making process of GNNs. The goal of these methods is to identify a small, label-preserving subgraph that retains the original model prediction $f(G)$ (Ying et al., 2019; Luo et al., 2020; Yuan et al., 2020; Wang & Shen, 2023). Due to the combinatorial search space, most existing approaches rely on continuous relaxation. They introduce a learnable mask $M$, applied to the input graph's adjacency matrix, and optimize $M$ by minimizing the discrepancy between the original prediction $f(G)$ and the masked prediction $f(M \odot G)$. The resulting soft masks are then binarized to yield the final discrete explanation subgraph (Yuan et al., 2021; Vu & Thai, 2020).

A key assumption underlying this post-hoc explanation framework is that the GNN model $f$, trained exclusively on unweighted graphs, will still produce meaningful predictions and reliable gradients when fed the continuously weighted graphs $M \odot G$ during mask optimization. This assumption is problematic as it introduces a significant distributional shift: the model is forced to process graphs with continuous edge weights, which deviate substantially from the binary-weighted graphs in its training distribution. This training scenario reflects the reality of many domains, such as molecular property prediction and social network analysis, where graph connections are often treated as binary (present or absent) due to data constraints or model simplification.

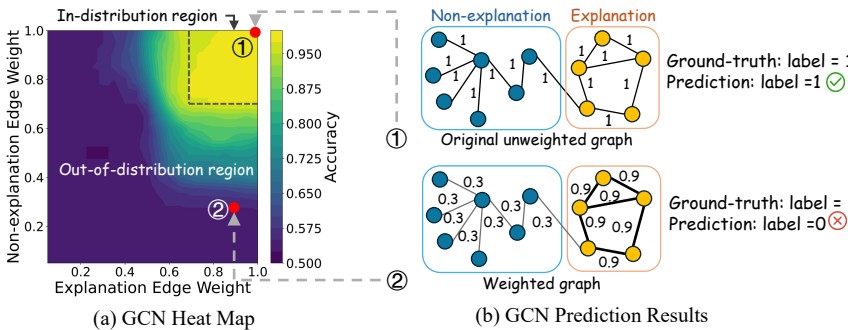

(a) GCN Heat Map      (b) GCN Prediction Results

Figure 1: (a) Classification accuracy of a GCN trained on unweighted graphs, evaluated on weighted graphs from the BA-2motif dataset with varying edge weights for motif and BA-graph edges. High accuracy is maintained only in the "in-distribution" region (both edge types near weight 1.0), but degrades significantly under distributional shift. (b) The original unweighted graph (example ①) is correctly classified, but reweighting motif edges to 0.9 and BA-graph edges to 0.3 (example ②) causes misclassification.

This distributional shift is particularly severe due to the sparsity objective necessary for obtaining faithful explanations. Meaningful explanations typically correspond to small, concise subgraphs, which requires the optimization to push most mask values in $M$ toward zero to satisfy size constraints. This process results in highly sparse, weighted graphs that bear little resemblance to the model's training data. Operating in this out-of-distribution regime can cause unstable model behavior and yield unreliable gradient signals, ultimately culminating in suboptimal or unfaithful explanations.

To empirically validate the impact of distributional shifts, we conduct an experiment on the BA-2motifs dataset, which contains graphs with either house or cycle motifs attached to Barabási–Albert (BA) base graphs (Albert & Barabási, 2002). As shown in Figure 1(a), the classification accuracy of a GCN trained on unweighted graphs degrades significantly when evaluated on weighted graphs where edge weights deviate from the training distribution. High accuracy is maintained only in the upper-right "in-distribution" region where both motif and base edge weights are near 1.0. Figure 1(b) illustrates a concrete failure case: the original unweighted graph is correctly classified (Example ①), but reweighting edges to reflect a typical explanation configuration (Example ②) leads to misclassification. This critical outcome demonstrates that model predictions fail to remain consistent under shifted conditions. If the importance of explanatory edges cannot dominate predictions in this out-of-distribution regime, the robustness and trustworthiness of explanations generated by existing methods become highly questionable.

To address this critical limitation, inspired by surrogate explainable AI literature (Ribeiro et al., 2016; Hooker et al., 2019), we propose explaining a **robust surrogate model** trained to handle continuous edge weights, whose explanations serve as a faithful proxy for those of the original model. We introduce a novel iterative framework called graph explanation with stochastic re-weighting (STORE) to build and explain this model. Specifically, our method proceeds through alternating phases of *subgraph identification* and *model adaptation*. Initially, we extract a relatively large explanation subgraph where soft mask optimization remains reliable. We then assign different weights to edges within and outside this subgraph and train surrogate models by adding these weighted graphs to the training datasets. This step enables surrogate models to maintain predictive consistency under the specific weight distribution that resembles soft-masked graphs, ensuring the validity of subsequent mask optimization. The process iterates with progressively smaller subgraphs, creating a "ladder"in which each step maintains the validity of the soft relaxation assumption. This framework offers several key advantages. First, it directly mitigates the distributional shift problem by ensuring that surrogate models can process weighted graphs at each stage of explanation refinement. Second, the progressive refinement from larger to smaller subgraphs provides a stable optimization trajectory, avoiding the severe distributional shifts that occur when directly searching for minimal explanations. Third, it naturally produces explanations at multiple levels of granularity, offering practitioners flexibility in balancing explanation size with fidelity. We summarize our contributions as follows:

- We identify and formally characterize a critical flaw in post-hoc GNN explainability: the distributional shift between unweighted training graphs and the weighted graphs used during mask optimization, which undermines explanation reliability.

- We propose a robust surrogate framework designed to resolve this OOD problem. By training a surrogate model capable of processing continuous edge weights, our framework ensures reliable gradient estimation and preserves the validity of soft mask optimization.
- We instantiate this framework through a novel iterative stochastic re-weighting strategy. This approach progressively aligns the model's training distribution with the explanation setting by assigning importance-aware weights to explanatory and non-explanatory edges.
- Extensive experiments on five benchmark datasets demonstrate that our method consistently improves explanation quality with different GNNs and explanation methods. while incurring only modest computational overhead.

## 2 PRELIMINARIES AND THE OOD PROBLEM

We represent a graph as $G = (\mathcal{V}, \mathcal{E}, \boldsymbol{X}, \boldsymbol{A})$, where $\mathcal{V} = \{v_1, v_2, \ldots, v_n\}$ is the set of nodes with $n = |\mathcal{V}|$, $\mathcal{E} \subseteq \mathcal{V} \times \mathcal{V}$ is the set of edges, $\boldsymbol{X} \in \mathbb{R}^{n \times d}$ is the node feature matrix where the $i$-th row $\boldsymbol{X}_i$ represents the $d$-dimensional feature of node $v_i$, and $\boldsymbol{A} \in \{0, 1\}^{n \times n}$ is the binary adjacency matrix such that $A_{i,j} = 1$ if $(v_i, v_j) \in \mathcal{E}$. Each graph is associated with a label $y \in \mathcal{Y}$.

Let $f : \mathcal{G} \mapsto \{1, 2, \cdots, |\mathcal{Y}|\}$ denote a to-be-explained GNN model that maps an input graph $G \in \mathcal{G}$ to its predicted class. This work focuses on *post-hoc, instance-level* explanation methods, which treat the GNN as a black box and aim to provide task-agnostic explanations for its predictions (Ying et al., 2019; Luo et al., 2020; Yuan et al., 2022; Huang et al., 2024). Formally, the objective of explaining a GNN model is to extract the most influential subgraph, $G_{exp} \subseteq G$, that contains the essential evidence for the model's prediction, $f(G)$. An ideal explanation adheres to two fundamental principles: *sufficiency* and *minimality*. Sufficiency gauges whether the explanation $G_{exp}$ is expressive enough on its own to yield the same prediction as the full graph. We can formalize this criterion by stating that the model's output distribution should not significantly diverge when conditioned on the subgraph versus the original graph. This is often quantified using the total variation distance: $d_{TV}(P_{\widehat{Y}|G}, P_{\widehat{Y}|G_{exp} \subseteq G}) \approx 0$. On the other hand, minimality serves as a counteracting principle, compelling the explanation to be as sparse as possible. This constraint, which typically involves minimizing the number of edges, ensures the explanation is easily interpretable. These two properties can be formalized in a single objective. The goal is to find an explainer function $\Psi(\cdot)$, where $G_{exp} = \Psi(G)$, that achieves the best possible sufficiency for a given minimality budget $s$:

$$\Psi^* = \arg\min_{\Psi} \mathbb{E}_G \left[ d_{TV}(f(G), P_{\widehat{Y}|G_{exp} \subseteq G}) \right] \quad \text{s.t.} \quad \mathbb{E}_G[|\mathcal{E}_{exp}|] \leq s. \tag{1}$$

Note that we have substituted the model's output $f(G)$ for the posterior $P_{\widehat{Y}|G}$. The primary challenge in solving equation 1 is the intractability of estimating the term $P_{\widehat{Y}|G_{exp} \subseteq G}$. Existing methods can be seen as providing different approximations to this optimization problem.

To practically solve the optimization problem in equation 1, existing methods learn a parameterized explainer function, denoted as $\Psi_\psi$, where $\psi$ represents the learnable parameters. For a given graph $G$, this function generates an intermediate, weighted subgraph, $\Psi_\psi(G)$, where edge weights signify their importance to the model's prediction. The final discrete explanation, $G_{exp}$, is then typically derived by binarizing this weighted subgraph.

The standard approach in the literature is to optimize the parameters $\psi$ by computing the sufficiency loss on this intermediate weighted subgraph. The formal objective from equation 1 is thus approximated by an objective over $\psi$. The sufficiency term is replaced by a proxy like the Cross-Entropy (CE) loss, and the minimality constraint is handled by a regularization term that encourages sparsity in the generated subgraph, such as the norm of its weighted adjacency matrix, $||\Psi_\psi(G)||$. This leads to the following standard optimization objective:

$$\psi^* = \arg\min_{\psi} \left( \sum_{(G,y) \in \mathcal{G}} \text{CE}(f(G), f(\Psi_\psi(G))) + \lambda ||\Psi_\psi(G)|| \right). \tag{2}$$

However, we claim this standard formulation contains a critical flaw. The GNN model $f$ is typically trained on graphs with unweighted edges (i.e., binary adjacency matrices). The intermediate subgraph, $\Psi_\psi(G)$, is inherently weighted. Consequently, evaluating the original model $f$ on this subgraph constitutes an OOD problem. The gradients produced in this regime, which are used to update the explainer's parameters $\psi$, may be unstable and unreliable.

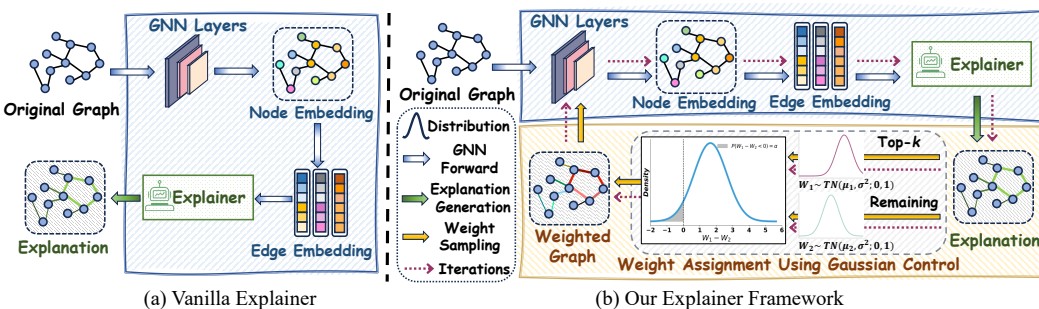

(a) Vanilla Explainer          (b) Our Explainer Framework

Figure 2: Overview of the vanilla explainer and STORE. (a) Vanilla explainer. (b) STORE first inputs the original graph into a GNN to obtain node and edge embeddings. Then we use an explainer to generate a weighted explanation subgraph. Based on this explanation, we apply a Gaussian-based sampling strategy to assign edge weights and construct a weighted graph. This graph is used to retrain a surrogate GNN to improve explanation accuracy.

## 3 ROBUST SURROGATE FRAMEWORK

In this section, we present our graph explanation with stochastic re-weighting (STORE) framework. An overview is shown in Figure 2(b). Our approach addresses the distributional shift problem by iteratively building a **robust surrogate model** through a process that alternates between *subgraph identification* and *model adaptation*. Given a pre-trained GNN model, we first extract a relatively large explanation subgraph where soft mask optimization remains reliable. We then construct weighted training graphs by assigning different weights to edges within and outside the identified subgraph, and retrain a surrogate GNN on these weighted graphs to ensure predictive consistency under the corresponding weight distribution. This process iterates for $L$ rounds, progressively refining explanations from larger to smaller subgraphs. At each iteration, the explainer is trained to identify important substructures, while the GNN learns to maintain consistent predictions on weighted graphs, thereby preserving the validity of continuous relaxation throughout the explanation refinement process.

### 3.1 EXPLANATION-GUIDED STOCHASTIC RE-WEIGHTING

Our iterative process begins by generating a foundational set of explanations to guide model adaptation. To accomplish this, we initialize $f^{(0)}$ with $f$, where $f$ is the pre-trained GNN model to be explained, and train an initial explainer, $\Psi_\psi^{(0)}$ on $f^{(0)}$ using the standard objective from equation 2. This process yields a continuous score for each edge, representing its learned importance. From these scores, we derive an initial explanation subgraph by selecting the top-$k$ edges. Following a common strategy, we use a relatively large '$k$' in this first step to create an inclusive set of potentially important edges.

This initial subgraph provides the necessary input for our re-weighting strategy. However, since the explanation is generated under the previously discussed OOD condition, the core of our method is to address this distributional shift directly. We achieve this by training a surrogate GNN model on weighted graphs constructed through a process we call **stochastic re-weighting**. This adaptation step supervises the surrogate model to maintain consistent predictions when processing graphs with varying edge weights, which is a critical requirement for generating reliable explanations.

As illustrated in Figure 2(b), given explanation subgraphs from the initial explainer, we construct weighted training graphs by stochastically assigning different weights to explanatory and non-explanatory edges indicated by the explanation subgraphs. Specifically, we use two well-separated truncated Gaussian distributions: explanatory edges receive weights sampled from a higher-mean distribution, while non-explanatory edges are sampled from a lower-mean distribution. This stochastic re-weighting strategy ensures that the surrogate model learns to respect the relative importance of explanatory edges, and the controlled randomness helps it generalize to the continuous weight variations encountered during soft mask optimization.

**Gaussian Parameter Selection.** Our stochastic re-weighting strategy samples edge weights to construct new weighted training graphs. Specifically, we use two truncated Gaussian distributions: explanatory edges receive weights sampled from $W_1 \sim \mathcal{TN}(\mu_1, \sigma^2; 0, 1)$, while non-explanatory

edges receive weights from $W_2 \sim \mathcal{TN}(\mu_2, \sigma^2; 0, 1)$. We use the $[0, 1]$ interval because this range directly mimics the continuous, weighted graphs generated by the soft mask optimization in standard GNN explainers, which is essential for bridging the distributional shift.

To parameterize these distributions, we first fix two hyperparameters: the desired mean separation $\Delta\mu = \mu_1 - \mu_2 > 0$ and the overlap probability $\alpha = \mathbb{P}(\tilde{W}_1 < \tilde{W}_2)$, where $\tilde{W}_1 \sim \mathcal{N}(\mu_1, \sigma^2)$ and $\tilde{W}_2 \sim \mathcal{N}(\mu_2, \sigma^2)$ are the untruncated counterparts of $W_1$ and $W_2$. We set $\alpha = 0.001$ to ensure a clear separation between the two edge types. The overlap probability is given by: $\mathbb{P}(\tilde{W}_1 < \tilde{W}_2) = \mathbb{P}(\tilde{W}_1 - \tilde{W}_2 < 0) = \Phi\left(\frac{-\Delta\mu}{\sqrt{2\sigma^2}}\right)$, where $\Phi(\cdot)$ is the standard normal cumulative distribution function: $\Phi(w) = \frac{1}{\sqrt{2\pi}} \int_{-\infty}^{w} e^{-t^2/2} \, dt$. By fixing $\Delta\mu$ and $\alpha$, the variance $\sigma^2$ is computed as: $\sigma^2 = \frac{(\Delta\mu)^2}{2[\Phi^{-1}(\alpha)]^2}$. Once $\sigma$ is determined, we select the means. We sample $\mu_2$ from $\text{Uniform}(2\sigma, 1 - \Delta\mu - 2\sigma)$, which in turn determines $\mu_1 = \mu_2 + \Delta\mu$. This sampling strategy ensures that both $\mu_1$ and $\mu_2$ are positioned at least $2\sigma$ from the 0 and 1 boundaries. Consequently, the probability of the untruncated Gaussians $\tilde{W}_1$ and $\tilde{W}_2$ falling outside $[0, 1]$ is negligible. This alignment ensures that the truncated distributions $\mathcal{TN}$ used in practice retain means and variances almost identical to their untruncated $\mathcal{N}$ counterparts, preserving the desired separation properties.

**Constructing Weighted Training Graphs.** For each graph $G$ in the training set, we use the current explainer $\Psi$ to identify the set of explanatory edges $\mathcal{E}^{\text{exp}}$. We then construct a weighted graph $G'$ by assigning edge weights sampled from $\mathcal{TN}(\mu_1, \sigma^2; 0, 1)$ to explanatory edges and from $\mathcal{TN}(\mu_2, \sigma^2; 0, 1)$ to non-explanatory edges. This stochastic re-weighting creates a training distribution that bridges the gap between unweighted graphs and the weighted graphs encountered during mask optimization. The full process is detailed in Appendix B Algorithm 2.

The surrogate GNN is then retrained on both the original graphs $\mathcal{G}$ and the weighted graphs $\mathcal{G}'$:

$$\arg\min_{f^{(l)}} \left( \sum_{(G,y) \in \mathcal{G} \cup \mathcal{G}'} -y \log(f^{(l)}(G)) \right), \tag{3}$$

This joint training ensures that the surrogate model maintains accurate predictions on both unweighted graphs (preserving original performance) and weighted graphs (enabling reliable mask optimization).

### 3.2 ITERATIVE REFINEMENT PROCESS

Our framework alternates between training the surrogate GNN on weighted graphs and updating the explainer, progressively refining explanations over $L$ iterations. After retraining the surrogate at each iteration, we update the explainer using the new model $f^{(l)}$. This iterative process, summarized in Appendix B Algorithm 1, progressively refines from larger to smaller subgraphs, which alleviates distributional shifts that occur when directly searching for minimal explanations.

**Remark.** Despite involving iterative retraining, the computational overhead remains modest. In practice, we find that the framework requires only a small number of iterations ($\leq 3$), and significant improvements are often observable after just a single adaptation step. As empirically demonstrated in Section 6.3 Figure 4, the cost per iteration is low, confirming that the iterative refinement process is computationally efficient.

## 4 THEORETICAL ANALYSIS

Our framework is designed to address the critical issue of distributional shift, where a GNN trained on unweighted graphs yields unreliable explanations when evaluated on the weighted graphs inherent to the explanation process. The iterative refinement process, which produces a final robust model, $f^{(L)}$, is empirically effective. Here, we provide the theoretical foundation that justifies when explanations from our final robust model can be considered faithful to the original, non-robust model, $f^{(0)}$.

Following Zheng et al. (2024), a task is $(s, \kappa)$-explainable if there exists an explanation function $\Psi$ such that the label is nearly conditionally independent of the full input given $\Psi(G)$, with expected explanation size at most $s$. A classifier is $(s, \zeta)$-explainable under the same condition with predictions replacing labels. We also adopt the stability assumption from Zheng et al. (2024), which requires that if $\Psi(g) \subseteq g'$, then $\Psi(g') = \Psi(g)$. This assumption ensures that explanations remain consistent when the input graph is extended to a supergraph. Under this uniqueness condition, we can formally

connect the explanations of the original model $f^{(0)}$ and the robust model $f^{(L)}$. The following theorem establishes that, when both models achieve errors close to the Bayes optimal, they share the same explanation function almost surely.

**Theorem 4.1** (Shared explanation for $f^{(0)}$ and $f^{(L)}$). *Consider a binary task with Bayes error $\epsilon^\star$. If the task is uniquely $(s, \kappa)$-explainable by $\Psi^\star$ under the stability assumption, and if $f^{(0)}$ and $f^{(L)}$ achieve errors at most $\epsilon^\star + \delta_0$ and $\epsilon^\star + \delta_L$, then both classifiers are $(s, \eta_i)$-explainable with respect to $\Psi^\star$ (with $\eta_i \to 0$ as $\delta_i, \epsilon^\star, \kappa \to 0$) and share the same explanation almost surely.*

This theorem shows that, under the stability and uniqueness assumptions, the robust model $f^{(L)}$ and the original model $f^{(0)}$ share the same underlying explanation almost surely. In other words, explanations from the robust model can be viewed as faithful surrogates for those of the original model, with the approximation error shrinking as both models approach Bayes optimality. The assumptions themselves are natural in many settings: stability reflects the idea that once a subgraph is sufficient to explain a prediction, adding irrelevant structure should not change the explanation; uniqueness means that for each instance, there is a well-defined explanatory subgraph. These principles echo the intuition behind widely used explanation methods such as LIME (Ribeiro et al., 2016), which rely on the existence of a consistent local explanation. Together, the assumptions highlight why our theoretical result is relevant in practice.

## 5 RELATED WORK

Existing research on GNN explainability primarily focuses on instance-level explanations, where the goal is to identify influential subgraphs for specific predictions. The dominant paradigm relies on perturbation, where methods optimize soft or binary masks to isolate subgraphs that preserve the original prediction. This category includes approaches that learn soft edge masks via optimization (Ying et al., 2019; Luo et al., 2020; Wang et al., 2021), employ search-based policies (Yuan et al., 2021; Li et al., 2023), or leverage game-theoretic values (Zhang et al., 2022). Despite improving fidelity, these methods inherently force the GNN to process modified graph structures that deviate from the discrete graphs seen during training, introducing artificial evidence.

To mitigate the OOD explanations, recent works such as MixupExplainer (Zhang et al., 2023) and ProxyExplainer (Chen et al., 2024) have proposed constructing proxy graphs or enforcing distribution-consistency constraints. However, these approaches overlook the specific disparity identified in our work where the target GNN is trained on unweighted graphs but explained using continuous weighted masks Unlike these methods, our STORE framework explicitly bridges this gap by training a robust surrogate model capable of handling the continuous edge weights inherent to the explanation process.

## 6 EXPERIMENTS

In this section, we conduct extensive experiments to evaluate the effectiveness of the proposed framework and compare it against widely used GNN explanation methods.

### 6.1 EXPERIMENTAL SETUP

**Datasets.** We evaluate our method on five benchmark datasets with ground-truth explanations. These include four real-world datasets: MUTAG (Luo et al., 2020), Benzene, Alkane-Carbonyl, and Fluoride-Carbonyl (Agarwal et al., 2023), as well as one synthetic dataset, BA-2motifs (Luo et al., 2020). Detailed dataset statistics and descriptions are provided in Appendix C.1.

**Baselines.** We compare our method with five representative explainers: GNNExplainer (Ying et al., 2019), PGExplainer (Luo et al., 2020), ReFine (Wang et al., 2021), MixupExplainer (Zhang et al., 2023), and ProxyExplainer (Chen et al., 2024). We follow the experimental settings in previous works (Ying et al., 2019; Luo et al., 2020) to train a three-layer Graph Convolutional Network (GCN) model (Kipf & Welling, 2017). Experiments on another representative GNN, Graph Isomorphism Network (GIN) (Xu et al., 2019), are provided in Appendix C.2.

**Settings.** Unlike previous works (Ying et al., 2019; Luo et al., 2020), which train a single GNN with a fixed seed and evaluate the explainer using 10 random seeds, we observe that the robustness of the GNN itself can vary substantially across different initializations and may exhibit different degrees of distributional shift. To obtain a more reliable and unbiased evaluation, we train both the GNN and the explainer under 10 random seeds. For each seed, a GNN is trained from scratch and an explainer is

Table 1: Performance comparison of different explanation methods using GCN as the backbone

| Explainer Method | BA-2motifs | MUTAG | Benzene | Alkane-Car. | Fluoride-Car. |
|---|---|---|---|---|---|
| GNNExplainer | $0.515_{\pm 0.107}$ | $0.707_{\pm 0.086}$ | $\mathbf{0.604}_{\pm 0.029}$ | $0.650_{\pm 0.042}$ | $0.640_{\pm 0.018}$ |
| + STORE | $\mathbf{0.528}_{\pm 0.108}$ | $\mathbf{0.720}_{\pm 0.057}$ | $0.595_{\pm 0.026}$ | $\mathbf{0.686}_{\pm 0.026}$ | $\mathbf{0.647}_{\pm 0.019}$ |
| PGExplainer | $0.672_{\pm 0.326}$ | $0.533_{\pm 0.272}$ | $0.541_{\pm 0.204}$ | $0.778_{\pm 0.237}$ | $0.784_{\pm 0.031}$ |
| + STORE | $\mathbf{0.758}_{\pm 0.232}$ | $\mathbf{0.774}_{\pm 0.189}$ | $\mathbf{0.662}_{\pm 0.152}$ | $\mathbf{0.801}_{\pm 0.227}$ | $\mathbf{0.802}_{\pm 0.020}$ |
| Refine | $0.522_{\pm 0.078}$ | $0.502_{\pm 0.193}$ | $0.723_{\pm 0.166}$ | $0.623_{\pm 0.268}$ | $0.633_{\pm 0.051}$ |
| + STORE | $\mathbf{0.546}_{\pm 0.095}$ | $\mathbf{0.529}_{\pm 0.197}$ | $\mathbf{0.747}_{\pm 0.178}$ | $\mathbf{0.752}_{\pm 0.153}$ | $\mathbf{0.648}_{\pm 0.043}$ |
| MixupExplainer | $0.847_{\pm 0.121}$ | $\mathbf{0.799}_{\pm 0.154}$ | $0.555_{\pm 0.270}$ | $0.444_{\pm 0.312}$ | $0.664_{\pm 0.090}$ |
| + STORE | $\mathbf{0.871}_{\pm 0.110}$ | $0.783_{\pm 0.160}$ | $\mathbf{0.570}_{\pm 0.252}$ | $\mathbf{0.493}_{\pm 0.299}$ | $\mathbf{0.667}_{\pm 0.130}$ |
| ProxyExplainer | $0.729_{\pm 0.226}$ | $0.614_{\pm 0.232}$ | $0.584_{\pm 0.214}$ | $0.678_{\pm 0.198}$ | $0.718_{\pm 0.036}$ |
| + STORE | $\mathbf{0.812}_{\pm 0.137}$ | $\mathbf{0.615}_{\pm 0.233}$ | $\mathbf{0.615}_{\pm 0.190}$ | $\mathbf{0.766}_{\pm 0.128}$ | $\mathbf{0.742}_{\pm 0.029}$ |

Table 2: Fidelity evaluation

| | | BA-2motifs | MUTAG | Fluoride-Carbonyl | D&D | Graph-SST2 |
|---|---|---|---|---|---|---|
| $Fid_{\alpha_1,+} \uparrow$ | PGExplainer | $0.038_{\pm 0.029}$ | $0.042_{\pm 0.008}$ | $0.055_{\pm 0.012}$ | $0.010_{\pm 0.011}$ | $0.008_{\pm 0.002}$ |
| | + STORE | $\mathbf{0.074}_{\pm 0.023}$ | $\mathbf{0.060}_{\pm 0.002}$ | $\mathbf{0.064}_{\pm 0.010}$ | $\mathbf{0.022}_{\pm 0.008}$ | $\mathbf{0.011}_{\pm 0.001}$ |
| $Fid_{\alpha_2,-} \downarrow$ | PGExplainer | $0.037_{\pm 0.023}$ | $0.063_{\pm 0.044}$ | $0.018_{\pm 0.007}$ | $0.003_{\pm 0.005}$ | $0.005_{\pm 0.002}$ |
| | + STORE | $\mathbf{0.035}_{\pm 0.022}$ | $\mathbf{0.041}_{\pm 0.022}$ | $\mathbf{0.016}_{\pm 0.008}$ | $\mathbf{0.001}_{\pm 0.004}$ | $\mathbf{0.003}_{\pm 0.002}$ |

applied to that model, and we report the final results as the average across the 10 runs. Additional implementation details are provided in Appendix C.3.

## 6.2 QUANTITATIVE EVALUATION

We report the mean AUC-ROC over 10 random runs to quantitatively evaluate the quality of explanations. Table 1 shows the results for five explanation methods, each evaluated in both their vanilla form and when integrated with our iterative framework, using GCN as the backbone. From the results, we have several key observations: First, although different explanation methods exhibit varying levels of performance, all of them consistently benefit from our iterative framework, STORE, which improves AUC-ROC scores over their vanilla counterparts across most datasets.

For example, on BA-2motifs, we observe a 12.8% relative gain for PGExplainer and a 11.4% improvement for ProxyExplainer, confirming the generalizability of our method across both synthetic and real-world datasets. Finally, STORE enhances stability, particularly on OOD-prone datasets such as BA-2motifs. While vanilla explainers (e.g., MixupExplainer) experience instability in some datasets, our method mitigates this by preserving semantically important substructures during iterative training. This targeted perturbation reduces the risk of degrading important graph structures, resulting in more reliable explanations. Moreover, Appendix D.1 reports results under the GIN architecture, showing that STORE is compatible with different GNN architectures and explainers.

To further evaluate the faithfulness of explanations, we adopt fidelity-based metrics. While standard fidelity metrics (Yuan et al., 2021) are widely used, they suffer from OOD problem (Amara et al., 2023), which compromise their reliability. To address this, we use the robust fidelity scores ($Fid_{\alpha_1,+}$, $Fid_{\alpha_2,-}$), following the formulation in (Zheng et al., 2024), with the default parameters $\alpha_1 = 0.1, \alpha_2 = 0.9$. As shown in Table 2, STORE consistently outperforms all baselines under both metrics, including the large-scale D&D (Dobson & Doig, 2003) and Graph-SST2 datasets (Yuan et al., 2022). We also provide a visual comparison in Appendix D.4, including the ground-truth explanations, those produced by the vanilla PGExplainer, and those generated by our method.

## 6.3 EFFICIENCY ANALYSIS

**Analysis of Iterations.** To determine whether only a small number of iterations is sufficient for stable performance gains, we vary the number of refinement iterations in STORE and evaluate the explanation results. We report AUC-ROC results for iteration counts ranging from 0 to 5. Additionally, we implement a progressively shrinking top-$k$ strategy during iterative retraining: at each iteration, only the top-$k$ most important edges are retained in the explanation graph, with $k$ decreasing from 0.9 to 0.1 across iterations 1 through 5. The progressively shrinking strategy begins with broad

subgraph exploration and incrementally focuses on discriminative structures. This process enforces finer attention to class-relevant structures while mitigating distractions from less informative edges.

As shown in Figure 3, on BA-2motifs, the AUC-ROC steadily increases during the first few iterations and plateaus after iteration 3, suggesting that iterative refinement and stronger substructure contrast effectively enhance explanation fidelity and model robustness. On MUTAG and Fluoride-Carbonyl, performance peaks at iteration 3 and slightly declines thereafter. This

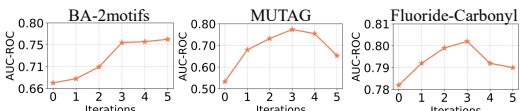

Figure 3: AUC-ROC performance with different numbers of iterations.

suggests that overly aggressive pruning of edges, resulting from smaller $k$ values in later iterations, may remove informative substructures, thus impairing both explanation fidelity and model generalization. Notably, a substantial improvement already appears at iteration 1 across all datasets compared with the original model at iteration 0, demonstrating that STORE does not rely on many refinement steps and showing that even one or two iterations are sufficient to achieve stable performance gains.

Moreover, the shrinking top-$k$ strategy encourages the explainer to focus on the most important edges over time. The observed improvements suggest that this gradual refinement enhances explanation accuracy and improves the robustness of the GNN to noise from less important structures. We also find that the performance after iterative retraining consistently surpasses the original baseline across all iterations, which suggests the robustness of our method.

**Running Time.** Figure 4 illustrates the computational efficiency of our framework by reporting the average runtime calculated over 10 independent runs. We compare the standard approach, which includes GCN training and PGExplainer optimization, against one complete iteration of STORE that involves stochastic graph construction and model retraining. The results indicate that the additional overhead is modest across all datasets. Notably, for BA-2motifs and Fluoride-Carbonyl, the runtime for one iteration is nearly equivalent to the standard training baseline, while the increase on MUTAG remains within a highly practical range. Given the significant improvements in explanation accuracy shown in Figure 3, these results confirm that our iterative framework achieves substantial performance gains with minimal computational cost.

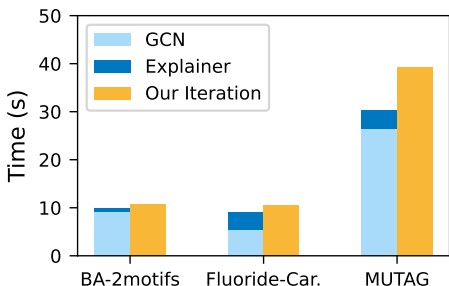

Figure 4: Average training time. GCN and Explainer jointly represent the standard pipeline.

### 6.4 ABLATION STUDIES

**Sampling Methods.** To validate our Gaussian-based re-weighting, we compare it against two baselines: i) Random Perturbation: Edge weights are sampled uniformly from $(0, 1)$, ignoring explainer importance. ii) Uniform-Based Sampling: A structured uniform approach where explanatory edges are sampled from $W_1 \sim \text{Uniform}(u, 1)$, and non-explanatory edges are scaled by noise $Z \sim \text{Uniform}(0, 1)$ such that $W_2 = W_1 \cdot Z$. This concentrates non-explanatory weights in $(0, u)$, providing separation but lacking the smooth distributional properties of our Gaussian approach.

Table 3: Performance comparison with different sampling methods

|  | BA-2motifs | MUTAG | Fluoride-Carbonyl |
|---|---|---|---|
| Random | $0.631_{\pm 0.282}$ | $0.513_{\pm 0.264}$ | $0.763_{\pm 0.035}$ |
| Uniform | $0.678_{\pm 0.241}$ | $0.757_{\pm 0.154}$ | $0.795_{\pm 0.022}$ |
| Gaussian | $0.758_{\pm 0.232}$ | $0.774_{\pm 0.189}$ | $0.802_{\pm 0.020}$ |

Table 3 reports the performance on BA-2motifs, MUTAG, and Fluoride-Carbonyl under different sampling strategies. We observe that the random perturbation yields the lowest AUC-ROC scores and the largest variances, indicating that random noise disrupts the learning process, leading to reduced stability. The Uniform-based strategy leads to notable performance gains compared to

random perturbation, especially on MUTAG and Fluoride-Carbonyl, demonstrating that incorporating edge importance helps preserve informative structures. The Gaussian-based strategy used in STORE achieves the best performance on all three datasets. It outperforms the other two variants in average performance, suggesting that it provides a better balance between edge importance preservation and controlled perturbation.

**Effects of** $\Delta\mu$**.** We conduct a hyperparameter analysis to investigate how the difference $\Delta\mu$ between the truncated Gaussian distributions for explanatory and non-explanatory edges affects the explainer's performance. As shown in Figure 5, we vary $\Delta\mu$ from 0.1 to 0.7 and report the AUC-ROC scores on BA-2motifs, MUTAG, and Fluoride-Carbonyl. We observe that the

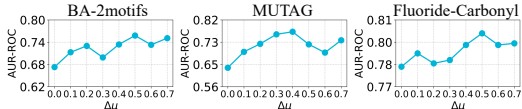

Figure 5: AUC-ROC performance with different $\Delta\mu$ for PGExplainer.

performance remains relatively stable across a wide range of $\Delta\mu$ values. On BA-2motifs and Fluoride-Carbonyl, the AUC-ROC improves steadily with increasing $\Delta\mu$ and reaches its peak around 0.5. On MUTAG, the performance improves initially and peaks at $\Delta\mu = 0.4$, then drops slightly beyond that. These findings indicate that a moderate separation between important and unimportant edges helps the model better distinguish informative substructures.

## 6.5 ALLEVIATING DISTRIBUTION SHIFTS

In this section, we visualize the classification accuracy under different edge weight configurations, where the x-axis denotes explanation edge weights and the y-axis denotes non-explanation edge weights. Figure 6 presents heat maps on BA-2motifs, MUTAG, and Fluoride-Carbonyl. For each dataset, we compare the original GCN model with our explanation-aware retrained model.

In the original model, we observe that only the upper-right region achieves high accuracy, where both explanation and non-explanation edge weights are large. In contrast, accuracy drops significantly in the lower-right region, where the graph contains only explanation edges. These results indicate that the model heavily relies on the global graph structure and fails to generalize when presented with explanation subgraphs, revealing the OOD problem. On the MUTAG dataset, we further find that the model can still achieve high accuracy even when explanation edge weights are small and non-explanation edge weights are large. This suggests that the model may exploit spurious background structures to make predictions, thereby undermining the faithfulness of explanation subgraphs. After applying explanation-aware iteration training, the model achieves consistently higher accuracy in a broader range of configurations, especially in the lower-right region. This indicates that the model has learned to better focus on explanation structures and that our method mitigates the OOD problem. In Appendix D.2 and Appendix D.2, we further report classification results and qualitative visualizations, confirming that the retrained GCN preserves both performance and explanatory consistency.

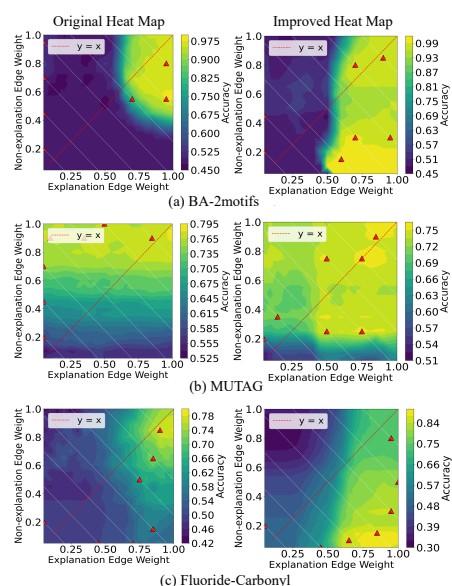

Figure 6: Heat maps of classification accuracy under different edge weights.

## 7 CONCLUSION

In this paper, we identify a critical flaw in post-hoc GNN explainability: the distributional shift caused by applying continuous explanation masks to models trained on unweighted graphs. To overcome this, we propose STORE, an iterative framework that constructs a robust surrogate model tailored to handle the weighted graphs inherent to the explanation process. By alternating between explanation identification and stochastic re-weighting, we ensure this surrogate serves as a faithful proxy for the original model while maintaining gradient reliability during mask optimization. Extensive experiments demonstrate that our method effectively mitigates the distribution mismatch, providing a general and model-agnostic solution for enhancing GNN interpretability.

## 8 REPRODUCIBILITY STATEMENT

We present all proofs of the main theorems in Appendix A and illustrate the overall workflow of our model in Appendix B using pseudocode. The source code required to reproduce our experiment is available at Anonymous Repository.

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

## A  THEORETICAL ANALYSIS: FULL PROOFS

This appendix provides the detailed definitions and proofs omitted from the main text. We follow the formalism of Zheng et al. (2024).

### A.1  PRELIMINARIES

**Definition A.1** (Explainable Classification Task (Zheng et al., 2024)). *A task with distribution $P_{Y,\boldsymbol{X},\mathbf{A}}$ is $(s,\kappa)$-explainable if there exists an explanation function $\Psi : \mathcal{G} \to 2^{\mathcal{V}} \times 2^{\mathcal{E}}$ such that*

$$I\big(Y;G \,\big|\, \mathbb{1}_{\Psi(G)}\big) \leq \kappa \quad and \quad \mathbb{E}_G\big[|\mathcal{E}_{\exp}|\big] \leq s,$$

*where $\Psi(G) = (\mathcal{V}_{\exp}, \mathcal{E}_{\exp})$.*

**Definition A.2** (Explainable Classifier (Zheng et al., 2024)). *For a classifier $f : \mathcal{G} \to \Delta_{\mathcal{Y}}$, let $\widehat{Y} \sim f(G)$. Then $f$ is $(s,\zeta)$-explainable if there exists $\Psi$ with*

$$I\big(\widehat{Y};G \,\big|\, \mathbb{1}_{\Psi(G)}\big) \leq \zeta \quad and \quad \mathbb{E}_G\big[|\mathcal{E}_{\exp}|\big] \leq s.$$

**Condition 1 (Stability under supergraphs).** For all $g, g'$, if $\Psi(g) \subseteq g'$, then $\Psi(g') = \Psi(g)$. This condition implies

$$I(\widehat{Y};G \,|\, \mathbb{1}_{\Psi(G)}) = I(\widehat{Y};G \,|\, \Psi(G)).$$

**Definition A.3** (Uniqueness of Task Explanation). *A binary task is uniquely $(s,\kappa)$-explainable if there exists an explanation $\Psi^\star$ satisfying Definition A.1 and Condition 1, and for any other valid $\Psi$ we have*

$$\mathbb{P}\big(\Psi(G) = \Psi^\star(G)\big) = 1.$$

### A.2  PROOF OF THEOREM 4.1

*Proof.* The Theorem 1 in Zheng et al. (2024) states that, if a task is $(s,\kappa)$-explainable with an explanation function satisfying Condition 1, then any classifier $f$ with error at most $\epsilon^\star + \delta$ is $(s,\eta)$-explainable for some $\eta \triangleq h_b(\tau) + \tau$, where $h_b(\tau) \triangleq -\tau \log \tau - (1-\tau) \log 1 - \tau$, $\tau \in [0,1]$ is the binary entropy function,

$$\tau \triangleq \frac{\big(2\sqrt{2}\epsilon^* + \sqrt{\xi}\big)^2}{2}, \quad \xi \triangleq \delta + e_{max}(\epsilon^*, \kappa) - \epsilon^*, \quad \delta \in (0, 9\epsilon^* - e_{max}(\epsilon^*, \kappa)),$$

and $e_{\max}(\epsilon^\star, \kappa)$ is the error bound from the reverse-Fano inequality, defined in the Proposition 1 of Zheng et al. (2024). In particular, $\eta \to 0$ as $(\delta, \epsilon^\star, \kappa) \to 0$. The classifiers $f^{(0)}$ and $f^{(L)}$ defined in our paper satisfy the accuracy conditions with error at most $\epsilon^\star + \delta$. Thus, both $f^{(0)}$ and $f^{(L)}$ are $(s, \eta_i)$-explainable with respect to certain explanation functions.

Furthermore, the task admits a unique explanation $\Psi^\star$ by using the fact that the explanation $\Psi^\star$ satisfies Definition A.1 and Condition 1 in Definition A.3. Thus, any valid explanation $\Psi$ that makes a classifier $(s, \eta)$-explainable must coincide with $\Psi^\star$ almost surely. Hence, both $f^{(0)}$ and $f^{(L)}$ are explained by the same $\Psi^\star$. The proof is completed. $\qquad\square$

### A.3  DISCUSSION

The proof highlights two essential assumptions:

1. Condition 1, which allows conditioning on $\Psi(G)$ directly.
2. Uniqueness of the task explanation, which ensures both classifiers align with the same $\Psi^\star$.

These strengthen the original statement and remove the gap caused by the possibility of multiple valid explanations.

# B ALGORITHM

---

**Algorithm 1** Iterative Explanation Framework

---

**Input:** A set of original graphs $\mathcal{G}$, GNN model $f(\cdot)$, explainer $\Psi(\cdot)$, number of iterations $L$
**Output:** explainer $\Psi^{(L)}(\cdot)$

1: $f^{(0)} \leftarrow f$
2: $\Psi^{(0)} \leftarrow$ TrainExplainer$(f^{(0)}, \mathcal{G})$
3: **for** $l = 1$ to $L$ **do**
4:     $\mathcal{G}^{\text{exp}} \leftarrow \Psi^{(l-1)}(\mathcal{G})$                                  $\triangleright$ Generate explanation graphs
5:     $\mathcal{G}' \leftarrow$ AugmentGraph$(\mathcal{G}, \mathcal{G}^{\text{exp}})$                      $\triangleright$ Construct augmentation
6:     $f^{(l)} \leftarrow$ TrainGNN$(\mathcal{G}')$                        $\triangleright$ Retrain GNN on augmented data
7:     $\Psi^{(l)} \leftarrow$ TrainExplainer$(f^{(l)}, \mathcal{G})$         $\triangleright$ Retrain explainer on original graphs
8: **end for**
9: **return** $\Psi^{(L)}(\cdot)$

---

**Algorithm 2** Construction of Augmented Graphs

---

**Input:** A set of original graphs $\mathcal{G}$, corresponding explanation graphs $\mathcal{G}^{\text{exp}}$, mean shift $\Delta\mu$, top-$k$ edges to select, separation probability $\alpha$
**Output:** A set of augmented graphs $\mathcal{G}'$

1: Compute variance: $\sigma^2 = \frac{(\Delta\mu)^2}{2(\Phi^{-1}(1-\alpha))^2}$
2: Sample $\mu_2 \sim \mathcal{U}(2\sigma, 1 - \Delta\mu - 2\sigma)$, set $\mu_1 = \mu_2 + \Delta\mu$
3: Initialize augmented graph set: $\mathcal{G}' \leftarrow \emptyset$
4: **for** $i = 1$ to $|\mathcal{G}|$ **do**
5:     Copy original graph: $\mathcal{G}^{\text{aug},i} \leftarrow \mathcal{G}^i$
6:     Select top-$k$ explanation edges: $\mathcal{E}^{\text{exp}} \leftarrow$ top-$k(\mathcal{G}^{\text{exp},i})$
7:     **for** each edge $e \in \mathcal{E}^{\text{exp}}$ **do**
8:         Sample $W_{1,e} \sim \mathcal{TN}(\mu_1, \sigma^2; 0, 1)$
9:         Assign $w(e) \leftarrow W_{1,e}$ in $\mathcal{G}^{\text{aug},i}$
10:    **end for**
11:    **for** each edge $e \in \mathcal{G}^i \setminus \mathcal{E}^{\text{exp}}$ **do**
12:       Sample $W_{2,e} \sim \mathcal{TN}(\mu_2, \sigma^2; 0, 1)$
13:       Assign $w(e) \leftarrow W_{2,e}$ in $\mathcal{G}^{\text{aug},i}$
14:    **end for**
15:    Add to augmented set: $\mathcal{G}' \leftarrow \mathcal{G}' \cup \{\mathcal{G}^{\text{aug},i}\}$
16: **end for**
17: **return** $\mathcal{G}'$

---

## C    DETAILED EXPERIMENTAL SETUP

### C.1    DATASETS

- **BA-2motifs** (Luo et al., 2020). This synthetic dataset contains 1,000 graphs generated from the Barabási–Albert (BA) model. Each graph belongs to one of two classes: one with house-shaped motifs and the other with five-node cycle structures.

- **MUTAG** (Luo et al., 2020). The MUTAG dataset includes 4,337 molecular graphs labeled according to their mutagenic effect on Salmonella typhimurium, a Gram-negative bacterium.

- **Benzene** (Agarwal et al., 2023). Benzene consists of 12,000 molecular graphs, categorized into two groups—those containing benzene rings and those without.

- **Alkane-Carbonyl** (Agarwal et al., 2023). This dataset contains 4,326 molecules. Positive samples are those with both alkane and carbonyl functional groups.

- **Fluoride-Carbonyl** (Agarwal et al., 2023). Fluoride-Carbonyl dataset includes 8,671 molecular graphs. The ground-truth explanation depends on the specific combination of fluoride atoms and carbonyl functional groups present in each molecule.

- **D&D** (Dobson & Doig, 2003). D&D consists of protein graphs labeled as either enzymes or non-enzymes. Nodes correspond to amino acids, and an edge is added between two nodes whenever their spatial distance is no greater than 6 Angstroms.

- **Graph-SST2** (Yuan et al., 2022). This dataset is a sentiment-analysis benchmark in which each sentence is converted into a graph representation, and the resulting graphs are annotated with their corresponding sentiment labels.

A summary of dataset statistics is provided in Table 4.

Table 4: Statistics of datasets used for graph classification task

| Dataset | Domain | #Graphs | Avg.#nodes | Avg.#edges | #Feature | #Classes |
|---|---|---|---|---|---|---|
| BA-2motifs | Synthetic | 1,000 | 25 | 51 | 10 | 2 |
| MUTAG | Biochemical molecules | 2,951 | 30.32 | 30.77 | 14 | 2 |
| Benzene | Biochemical molecules | 12,000 | 20.58 | 43.65 | 14 | 2 |
| Alkane-Car. | Biochemical molecules | 4,326 | 21.13 | 44.95 | 14 | 2 |
| Fluoride-Car. | Biochemical molecules | 8,671 | 21.36 | 45.37 | 14 | 2 |
| D&D | Bioinformatics | 1,178 | 284.32 | 715.66 | 89 | 2 |
| Graph-SST2 | Natural language | 70, 042 | 10.20 | 18.40 | 768 | 2 |

### C.2    BASELINES

- **GNNExplainer** (Ying et al., 2019). GNNExplainer learns soft masks over edges and node features to identify the most relevant substructures for a given prediction. These masks are applied to the original graph via element-wise multiplication and optimized by maximizing the mutual information between the prediction on the masked graph and the original graph.

- **PGExplainer** (Luo et al., 2020). This method extends the idea of GNNExplainer by assuming that the graph is a random Gilbert graph. It generates edge embeddings by concatenating the node embeddings and uses them to parameterize a Bernoulli distribution that determines whether an edge is included. A Gumbel-Softmax trick is applied for differentiable sampling during end-to-end training.

- **ReFine** (Wang et al., 2021). ReFine first learns class-wise edge probabilities by optimizing mutual information and a contrastive loss across classes. During fine-tuning, it generates instance-specific explanations by sampling edges based on the learned probabilities, aiming to maximize mutual information with the model's prediction.

- **MixupExplainer** (Zhang et al., 2023). MixupExplainer constructs explanations by mixing explanation subgraphs with randomly sampled, label-independent base graphs. This non-parametric augmentation strategy alleviates OOD issue commonly observed in previous explainers.

- **ProxyExplainer** (Chen et al., 2024). ProxyExplainer mitigates the distributional shift in explanation by generating in-distribution proxy graphs via a parametric graph generator, ensuring both fidelity and alignment with the original graphs.

## C.3 IMPLEMENTATION

We use the Adam optimizer (Kingma & Ba, 2014) with a learning rate of $1 \times 10^{-3}$ for training the GNN, and use the same optimizer with a weight decay of $5 \times 10^{-4}$ for training the explainer. The AUC-ROC metric is adopted for quantitative evaluation. All experiments are conducted on a Linux machine equipped with four NVIDIA A100-PCIE GPUs, each with 40 GB of memory. The CUDA version is 12.3, and the driver version is 545.23.08.

# D EXTRA EXPERIMENTAL STUDY

## D.1 EXPERIMENTS WITH GIN MODEL

In Table 5, we report results using GIN as the backbone architecture. The results show that STORE consistently improves the performance of various explanation methods across most datasets. For example, PGExplainer achieves a 29.2% improvement on Benzene and a 7.7% gain on Fluoride-Carbonyl, while ReFine and ProxyExplainer also show improvements on BA-2motifs and Fluoride-Carbonyl, respectively. These results further demonstrate the model-agnostic nature of our framework: even when switching to a different GNN architecture, the retraining process still improves explanation quality.

Table 5: Performance comparison of different explanation methods using GIN as the backbone

| Explainer Method | BA-2motifs | MUTAG | Benzene | Alkane-Car. | Fluoride-Car. |
|---|---|---|---|---|---|
| GNNExplainer | $0.511_{\pm 0.005}$ | $0.553_{\pm 0.127}$ | $0.469_{\pm 0.079}$ | $0.502_{\pm 0.126}$ | $0.501_{\pm 0.029}$ |
| + STORE | $\mathbf{0.544}_{\pm 0.053}$ | $\mathbf{0.589}_{\pm 0.114}$ | $\mathbf{0.514}_{\pm 0.018}$ | $\mathbf{0.552}_{\pm 0.071}$ | $\mathbf{0.513}_{\pm 0.005}$ |
| PGExplainer | $0.756_{\pm 0.315}$ | $0.489_{\pm 0.246}$ | $0.617_{\pm 0.260}$ | $0.495_{\pm 0.259}$ | $0.678_{\pm 0.088}$ |
| + STORE | $\mathbf{0.791}_{\pm 0.176}$ | $\mathbf{0.657}_{\pm 0.209}$ | $\mathbf{0.797}_{\pm 0.108}$ | $\mathbf{0.631}_{\pm 0.258}$ | $\mathbf{0.730}_{\pm 0.118}$ |
| Refine | $0.518_{\pm 0.066}$ | $0.535_{\pm 0.225}$ | $0.499_{\pm 0.127}$ | $0.603_{\pm 0.190}$ | $0.551_{\pm 0.074}$ |
| + STORE | $\mathbf{0.584}_{\pm 0.066}$ | $\mathbf{0.537}_{\pm 0.150}$ | $\mathbf{0.638}_{\pm 0.158}$ | $\mathbf{0.650}_{\pm 0.117}$ | $\mathbf{0.597}_{\pm 0.054}$ |
| MixupExplainer | $0.738_{\pm 0.281}$ | $\mathbf{0.480}_{\pm 0.178}$ | $0.593_{\pm 0.227}$ | $0.420_{\pm 0.260}$ | $0.767_{\pm 0.088}$ |
| + STORE | $\mathbf{0.759}_{\pm 0.161}$ | $0.475_{\pm 0.192}$ | $\mathbf{0.649}_{\pm 0.213}$ | $\mathbf{0.464}_{\pm 0.236}$ | $\mathbf{0.785}_{\pm 0.065}$ |
| ProxyExplainer | $0.621_{\pm 0.205}$ | $0.567_{\pm 0.110}$ | $0.537_{\pm 0.139}$ | $0.465_{\pm 0.156}$ | $\mathbf{0.605}_{\pm 0.055}$ |
| + STORE | $\mathbf{0.659}_{\pm 0.173}$ | $\mathbf{0.644}_{\pm 0.131}$ | $\mathbf{0.582}_{\pm 0.136}$ | $\mathbf{0.582}_{\pm 0.129}$ | $0.600_{\pm 0.067}$ |

## D.2 VALIDITY OF ROBUST SURROGATE MODEL

In the Section 6.2, we have demonstrated the effectiveness of STORE in generating high-quality explanations. These results confirm the practical utility of addressing the distributional shift. To further validate the consistency of our framework, specifically that the robust surrogate model remains a faithful proxy for the original model, we directly compare their behaviors. In this section, we assess the alignment between the two models across two critical dimensions, including predictive consistency and structural preservation.

**Predictive Consistency.** First, we examine whether the robust surrogate preserves the predictive capability of the original model. Table 6 reports the graph classification performance of the original GCN versus the final robust surrogate on the original unweighted test set. For MUTAG, we observe slight improvements in all metrics, while for BA-2motifs and Fluoride-Carbonyl, the performance remains highly stable. These findings indicate that while the surrogate model extends its domain to handle weighted graphs, it retains the predictive ability decision of the original model.

Table 6: Graph classification performance of the GCN model before and after iterations

|  |  | BA-2motifs | MUTAG | Fluoride-Carbonyl |
|---|---|---|---|---|
| Train Acc | Ori | $0.923_{\pm0.061}$ | $0.787_{\pm0.026}$ | $0.764_{\pm0.036}$ |
|  | Update | $0.915_{\pm0.057}$ | $0.802_{\pm0.010}$ | $0.757_{\pm0.038}$ |
| Val Acc | Ori | $0.942_{\pm0.050}$ | $0.785_{\pm0.025}$ | $0.773_{\pm0.444}$ |
|  | Update | $0.933_{\pm0.062}$ | $0.791_{\pm0.011}$ | $0.766_{\pm0.033}$ |
| Test Acc | Ori | $0.934_{\pm0.049}$ | $0.771_{\pm0.026}$ | $0.750_{\pm0.053}$ |
|  | Update | $0.926_{\pm0.062}$ | $0.786_{\pm0.014}$ | $0.752_{\pm0.040}$ |

**Consistency of Explanation Structures.** To further ensure that the surrogate model relies on the same reasoning evidence as the original model, we directly visualize the edge attributions produced by both models. We apply Grad to both the original and robust GCNs using the same input graphs. As shown in Figure 7, the highlighted substructures are highly consistent across both models, with only minor variations in intensity. This consistency confirms that the surrogate model preserves the reasoning logic of the original model, ensuring that the explanations generated via the surrogate are valid for the original target.

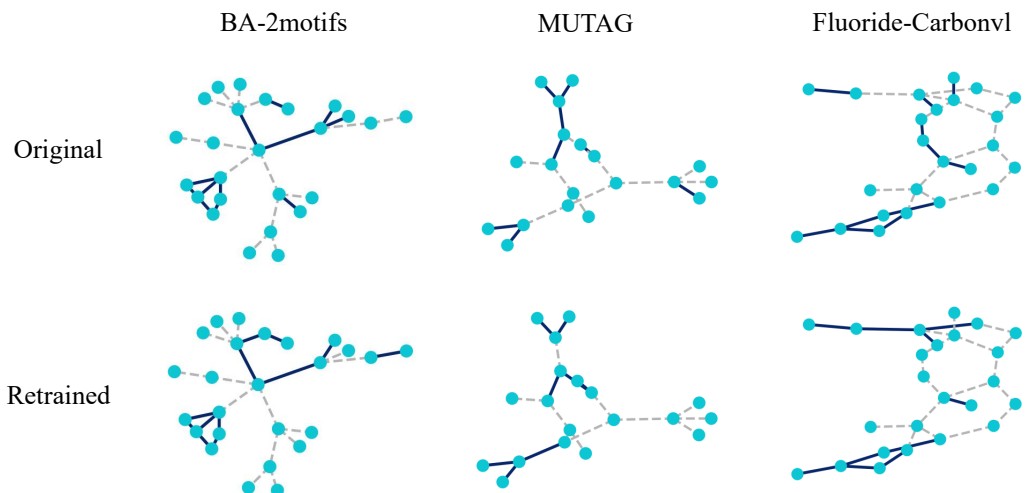

Figure 7: The explanation visualization comparison between original and retrained GCNs.

### D.3 Weight Distribution Visualization

To illustrate the edge weight sampling strategy used in our iterative framework, we visualize the distributions of sampled weights for explanatory and non-explanatory edges. Specifically, we draw samples from two truncated Gaussian distributions: $W_1 \sim \mathcal{TN}(\mu_1, \sigma^2; 0, 1)$ for important edges, and $W_2 \sim \mathcal{TN}(\mu_2, \sigma^2; 0, 1)$ for less important edges, where $\mu_1 > \mu_2$. Figure 8 presents the distributions of sampled edge weights for the BA-2motifs, MUTAG, and Fluoride-Carbonyl datasets. In each subfigure, the blue histogram represents the distribution of weights assigned to non-explanatory edges, and the green histogram corresponds to explanatory edges. We also overlay the histograms with kernel density estimation curves corresponding to the Gaussian distributions $\mathcal{N}(\mu_1, \sigma^2)$ and $\mathcal{N}(\mu_2, \sigma^2)$, shown as solid purple and dashed red curves, respectively.

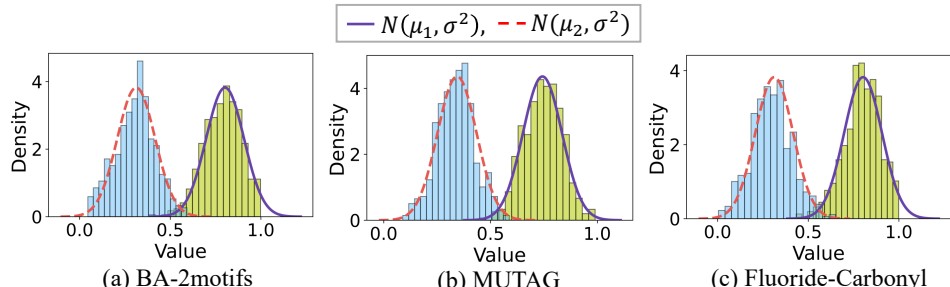

(a) BA-2motifs      (b) MUTAG      (c) Fluoride-Carbonyl

Figure 8: Sampled edge weight distributions for explanatory and non-explanatory edges.

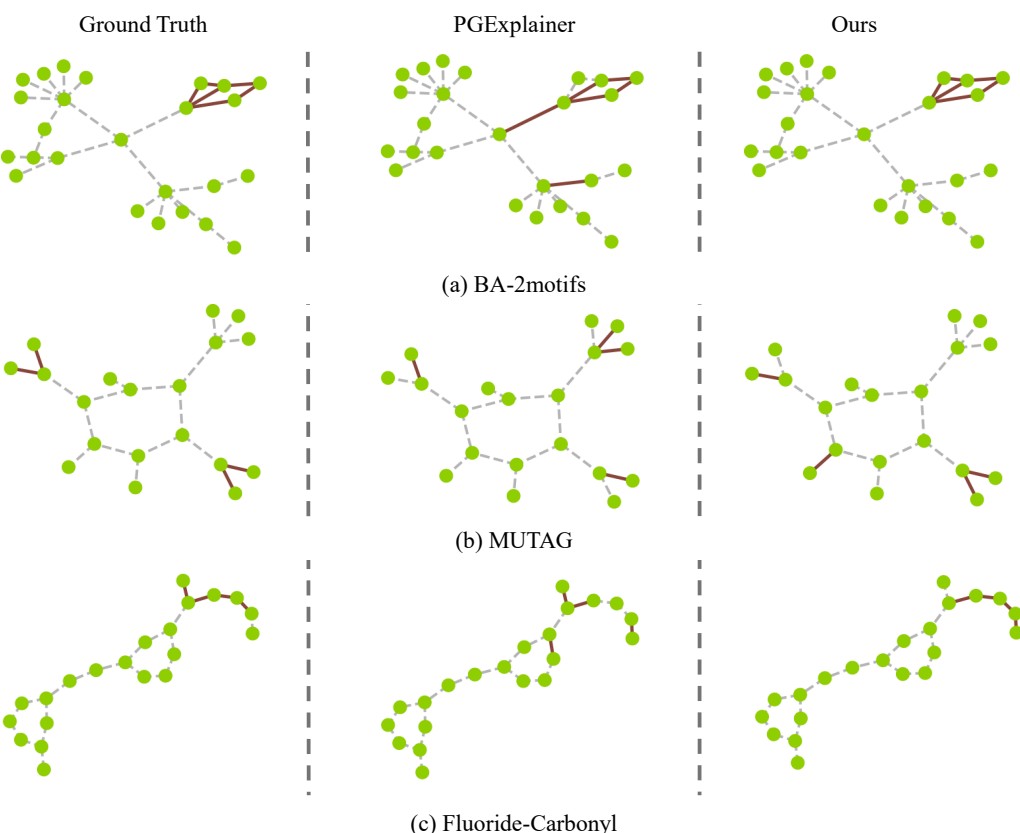

Figure 9: Visual comparison of explanation results.

As shown in the figure, while the actual sampling process is based on truncated Gaussian distributions, the curves demonstrate that the sampled histograms closely resemble the Gaussian distributions. In addition, the two distributions exhibit a clear separation. These results confirm that the Gaussian-based sampling strategy enhances structural signals through explanation-aware weighting, and constraints noise from irrelevant edges, enabling the GNN to learn more effectively.

### D.4 CASE STUDY

To qualitatively evaluate the effectiveness of STORE, we conduct case studies on representative samples. Figure 9 visualizes the ground-truth substructures, the explanations produced by the vanilla model, and those generated by our method. In each subfigure, important edges are highlighted in bold brown. For fair comparison, we select the same number of top-ranked edges from the explanation as the number of edges in the ground-truth. From the results, our method produces more focused and

faithful explanations compared to the vanilla baseline. Specifically, STORE consistently highlights the core discriminative substructures without introducing irrelevant edges, while the vanilla model often includes additional visualizations underscore STORE ability to provide concise subgraph explanations that align with the true decision-making process.

## E  USE OF LLMS

In this paper, we use Large Language Models (LLMs) to help improve the clarity and readability of the manuscript, including correcting typos and refining grammar.

