# OpenReview forum: "From Binary to Continuous: Stochastic Re-Weighting for Robust Graph Explanation"
_ICLR.cc/2026/Conference — Submitted to ICLR 2026_

### Official Review · Reviewer_xuGQ · 2025-10-29

**Soundness:** 3
**Presentation:** 3
**Contribution:** 2
**Rating:** 4
**Confidence:** 5

**Summary:**

This paper proposes STORE, an iterative framework to address the distributional discrepancy between GNN training and post-hoc explanation. After training a base GNN and an explainer, STORE retrains the GNN using structure-regularized graphs generated by the explainer, aligning the two distributions over several iterations. Experiments on synthetic and molecular graph datasets show improved explanation fidelity and stability.

**Strengths:**

- The motivation is clear and well articulated, highlighting an important gap in existing GNN explainers.
- The idea of iteratively aligning GNN and explainer training is interesting and may inspire follow-up work.
- The experiments are reasonably comprehensive, including both synthetic and real-world datasets, and Appendix I provides a runtime analysis suggesting modest overhead on small datasets.

**Weaknesses:**

- The proposed framework is unnecessarily heavy and complicated. To address the distributional discrepancy, STORE introduces a third iterative step that requires retraining both the GNN and the explainer, which significantly increases methodological and computational complexity.

- The paper does not convincingly justify why such retraining is essential. The same goal could likely be achieved with simpler mechanisms, for example a reinforcement learning–based approach that adds or modifies edges sequentially to align distributions without full retraining.

- While Section 5 provides a theoretical justification that retraining preserves explanation faithfulness, it does not analyze convergence behavior or guarantee that the iterative process stabilizes in practice.

- The efficiency study is limited to small datasets; there is no large-scale or FLOP-level analysis to demonstrate scalability.

- Although explanation fidelity improves, there is no clear evidence that retraining does not distort or degrade the base GNN’s performance on its original task.

**Questions:**

- How sensitive is STORE to the number of retraining iterations? Does performance plateau quickly?
- Can the authors clarify runtime or memory scaling on larger datasets?
- Does retraining the GNN affect its original classification accuracy?
- Could simpler alternatives, such as reinforcement learning for subgraph generation, achieve similar alignment without retraining?

---

> ### Author Response · Authors · 2025-11-20
> **Response to reviewer xuGQ (part 1/2)**
>
> We thank the reviewer for the helpful suggestions.
>
> > **W1.** The proposed framework is unnecessarily heavy and complicated. To address the distributional discrepancy, STORE introduces a third iterative step that requires retraining both the GNN and the explainer, which significantly increases methodological and computational complexity.
>
> Thank you for raising this point. While STORE introduces an iterative refinement step, the additional computational cost is modest. To address this concern, we have provided a detailed training time analysis in Appendix I (now moved to Sec.6.3). As shown in the following table, the cost of one complete iteration is small compared to the standard approach setup across three datasets.
>
> Furthermore, Sec.6.3 shows that we only need very few iterations (<3) to achieve the optimal performance and even a single iteration already has noticeable improvements in explanation accuracy. This indicates that STORE does not require many iterations to be effective, and the additional refinement step provides clear benefits with marginal overhead.
>
> | Dataset | Standard Approach (s) | Our Iteration (s) |
> |---------|------------------|-------------------------|
> | BA-2motifs | 9.92  | 10.67 |
> | MUTAG      | 30.38 | 39.17 |
> | Fluoride-Carbonyl | 9.10 | 10.50 |
>
> > **W2.** The paper does not convincingly justify why such retraining is essential. The same goal could likely be achieved with simpler mechanisms, for example a reinforcement learning–based approach that adds or modifies edges sequentially to align distributions without full retraining. (**Q4.** Could simpler alternatives, such as reinforcement learning for subgraph generation, achieve similar alignment without retraining?)
>
> Thank you for this insightful point. Our empirical observation reveals a previously overlooked issue: soft-mask explanations create weighted graphs that fall outside the training distribution of the original GNN. The iterative retraining step in STORE directly addresses this distributional mismatch by gradually adapting the model to weighted graphs. Importantly, STORE reuses existing GNN and explainer modules, keeping the design conceptually simple. Both our theoretical analysis and experimental results show that this retraining procedure is effective and lightweight.
>
> We thank the reviewer's the suggestion of using a reinforcement learning–based mechanism. It is an interesting idea. We would like to clarify that our main contributions are identifying an long-lasting overlooked problem in the explainable GNN field. and We provide a novel robust surrogate framework. Based on that, we provide an efficient and straightforward implementation. We are happy to provide our thought on other options. Specifically, we respectfully believe that RL-based implementation is non-trivial in this setting. It is unclear how to design (i) a reward function that faithfully captures distributional alignment, and (ii) an action space that sequentially modifies edges or weights in a stable and scalable manner. More importantly, RL-based approaches would likely introduce additional architectural and optimization complexity, whereas STORE avoids heavy designs and achieves high empirical performance with a straightforward strategy.
> We consider this an appealing direction for future work and plan to investigate it further.
>
> > **W3.** While Section 5 provides a theoretical justification that retraining preserves explanation faithfulness, it does not analyze convergence behavior or guarantee that the iterative process stabilizes in practice. (**Q1.** How sensitive is STORE to the number of retraining iterations? Does performance plateau quickly?)
>
> We thank the reviewer for the question. We provide an empirical analysis of iteration behavior in Sec.6.3. As shown in Figure 3, across three datasets, a substantial improvement already appears at iteration 1 compared to iteration 0, demonstrating that the framework achieves noticeable gains with only a single refinement step. Then  the performance continues to improve up to around iteration 3 and plateaus, indicating that the iterative process stabilizes quickly in practice.
> We also observe a slight decrease beyond iteration 3 on some datasets. Since our shrinking top-k strategy retains fewer edges in later iterations, overly aggressive pruning may remove certain informative substructures, leading to minor fluctuations. So in practice, we set the iteration to 3 and achieves consistent improvements.

---

> ### Author Response · Authors · 2025-11-20
> **Response to reviewer xuGQ (part 2/2)**
>
> > **W4.** The efficiency study is limited to small datasets; there is no large-scale or FLOP-level analysis to demonstrate scalability.
>
> We thank the reviewer for highlighting the importance of scalability. To further evaluate the scalability of our method, we additionally include experiments on two larger benchmarks: D&D and Graph-SST2. The D&D dataset contains graphs with an average of 284 nodes and 716 edges, while Graph-SST2 consists of 70,042 graphs. Since D&D and Graph-SST2 do not include ground-truth explanations, we report the fidelity metrics Fid α₁,+ and Fid α₂,− in the table below. STORE consistently improves explanation quality on these two large-scale dataset, demonstrating the practical scalability of our framework. We have added these results to the main text for completeness.
>
> | Data        | Explainer Method |    Fid α1,+ ↑   |    Fid α2,− ↓   |
> |-------------|------------------|-----------------|-----------------|
> | D&D         |    PGExplainer   |   0.010±0.011   |   0.003±0.005   |
> |             |      + STORE     | **0.022±0.008** | **0.001±0.004** |
> | Graph-SST2  |    PGExplainer   |   0.008±0.002   |   0.005±0.002   |
> |             |      + STORE     | **0.011±0.001** | **0.003±0.002** |
>
> > **W5.** Although explanation fidelity improves, there is no clear evidence that retraining does not distort or degrade the base GNN’s performance on its original task. (**Q3.** Does retraining the GNN affect its original classification accuracy?)
>
> Thanks for pointing this out. We have shown the effect of retraining on the base GNN's predictive performance in Appendix D.2 (previously, Appendix G). As shown in Table 6, the classification accuracy of the GNN before (Ori) and after (Update) the iterative refinement remains stable. These results demonstrate that the iterative retraining procedure does not distort or degrade the original GNN's classification ability. The predictive performance is well preserved throughout the process.
>
> > **Q2.** Can the authors clarify runtime or memory scaling on larger datasets?
>
> We appreciate the reviewer’s question regarding runtime on larger datasets. To provide a clear assessment, we evaluate STORE on three larger benchmarks, Benzene, D&D, and Graph-SST2. The runtime results are shown in the table below. These results indicate that STORE maintains a modest runtime compared to the standard approach setup, even on datasets with larger graph sizes.
>
> |     Dataset    | Standard Approach (s) | Our Iteration (s) |
> |----------------|------------------|------------------------|
> | Benzene        | 13.79 | 19.01 |
> | D&D            | 9.07  | 13.15 |
> | Graph-SST2     | 28.04 | 37.81 |

---

### Official Review · Reviewer_aHyN · 2025-10-29

**Soundness:** 1
**Presentation:** 2
**Contribution:** 1
**Rating:** 0
**Confidence:** 4

**Summary:**

The proposed STORE aims to provide more robust and reliable explanations by addressing the problem of explanatory graphs falling into out-of-distribution regions that are distant from the distribution of graphs in the training data.

**Strengths:**

The paper addresses an interesting research topic, handling out-of-distribution (OOD) issues in explanatory graphs, which has emerged as one of the most actively researched areas in the Graph XAI field.

**Weaknesses:**

1. Pre-trained GNNs should be treated as fixed targets for explanation. However, since the proposed method re-trains the target model to address the OOD explanatory graphs problem, the resulting explanations cannot be considered explanations of the original pre-trained GNN.
2. The approach requires retraining a pre-trained GNN, which represents a highly restrictive setting with limited feasibility in real-world applications where models need to remain fixed.
3. When the explanation fails to capture the ground-truth properly following the proposed STORE, the OOD problem may be addressed, but the performance after retraining will deteriorate significantly, and consequently, the explanation will also be learned in a more negative direction.
4. The iterative explanation framework does not account for computational cost and time requirements, while the gains are incremental.

**Questions:**

1. When the target GNN is retrained, the explanation target changes. If so, the provided explanation is not for the original GNN that we intended to explain. Then how can explanations for the original GNN be provided?
2. The original trained GNN and the retrained GNN are not theoretically and practically identical models. What is the rationale for providing explanations for the retrained GNN instead of the original trained GNN?
3. What are the performance changes of the GNN before and after retraining? Since it is trained on a dataset that includes explanatory graphs, what kind of performance changes occur?
4. In most papers, the weighted graph in Figure 1(b) is located in the OOD region, but unlike the example shown, the prediction label is not incorrect. This appears to be an extreme case if the given GNN is well-trained.

Please address the questions with consideration of the mentioned weaknesses.

---

> ### Author Response · Authors · 2025-11-20
> **Response to reviewer aHyN (part 1/2)**
>
> We appreciate the reviewer’s feedback.
>
> > **W1.** Pre-trained GNNs should be treated as fixed targets for explanation. However, since the proposed method re-trains the target model to address the OOD explanatory graphs problem, the resulting explanations cannot be considered explanations of the original pre-trained GNN. (**Q1.** When the target GNN is retrained, the explanation target changes. If so, the provided explanation is not for the original GNN that we intended to explain. Then how can explanations for the original GNN be provided?)
> (**Q2.** The original trained GNN and the retrained GNN are not theoretically and practically identical models. What is the rationale for providing explanations for the retrained GNN instead of the original trained GNN?)
>
> We appreciate this insightful comment. We agree that post-hoc explanations should reflect the behavior of the original pre-trained GNN. We show that directly optimizing soft masks on the original model is suboptimal due to the Out-Of-Distribution (OOD) shift. The original model, trained on binary graphs, produces unreliable gradients when fed the continuous (weighted) graphs required for mask optimization.
> To address this long-dismissed problem, we construct a **robust surrogate** that serves as a faithful proxy. This surrogate aligns with the original model on the in-distribution data (binary graphs) but provides reliable gradients in the continuous space. In our original submission, we provided rigorous verification of this faithfulness across theorical anaysis (Previous section 5), emprical study (section 6.4) and behavioral verification (Appendix G and H).  We have take this opportunity to polish and reorganize these sections. Specifically,
>
> **1. Theoretical Justification (Section 4)**
> We formalize the relationship between the original and retrained models. Theorem 4.1 establishes that if both the original model $f^{(0)}$ and the robust model $f^{(L)}$ achieve errors close to the Bayes optimal, they share the same explanation function almost surely. This holds under the standard stability and uniqueness assumptions, providing theoretically jusitication of our surroagte framework.
>
> **2. Behavioral Consistency (Appendix D.2)**
> We empirically verified that the retrained model preserves the behavior of the original pre-trained GNN:
> * **Predictive Consistency:** As shown in **Table 6**, the classification accuracy of the GCN before and after retraining is nearly identical (e.g., MUTAG Test Acc: 0.771 vs. 0.786; Fluoride-Carbonyl: 0.750 vs. 0.752).
>
> * **Structural Preservation :** We visualized the edge attributions (gradients) of both the original and retrained models in **Figure 7**. The highlighted substructures are highly consistent, confirming that the retraining process enhances robustness to weighted graphs without altering the model's focus on key structures.
>
> **3. Explanation Quality (Section 6.2)**
> We would like to respecfully clarify that our evaluation framework, AUC-ROC, and Fidelity are based on the original model. **Table 1** shows that our method consistently improves AUC-ROC (matching ground truth), and **Table 2** demonstrates superior Fidelity scores. This indicates that our surrogate method recovers the underlying decision logic of the original GNN more accurately than baselines, which suffer from gradient noise caused by the OOD shift.
>
> > **W2.** The approach requires retraining a pre-trained GNN, which represents a highly restrictive setting with limited feasibility in real-world applications where models need to remain fixed.
>
> We appreciate the reviewer's perspective on the practical constraints of retraining. We would like to provide a perspective on why this setting aligns with common practices in XAI research, particularly when the goal is for model owners to debug and understand deep learning behaviors. In such contexts, access to the training architecture and data is standard and typically assumed.
>
> This alignment is reflected in foundational works in the field:
> * **LIME (Ribeiro et al., 2016):** A seminal work (over 27k citations) which relies on sampling from the input space to train local surrogates, implicitly operating under the assumption that the data distribution is accessible.
> * **ROAR (Hooker et al., 2019):** A standard evaluation benchmark (Remove and Retrain, ~1k citations) which explicitly necessitates iteratively retraining the model on modified datasets to ensure rigorous fidelity measurement.
>
> Thus, we believe that our framework operates within established conventions for high-fidelity XAI.

---

> ### Author Response · Authors · 2025-11-20
> **Response to reviewer aHyN (part 2/2)**
>
> > **W3.** When the explanation fails to capture the ground-truth properly following the proposed STORE, the OOD problem may be addressed, but the performance after retraining will deteriorate significantly, and consequently, the explanation will also be learned in a more negative direction. (**Q3.** What are the performance changes of the GNN before and after retraining? Since it is trained on a dataset that includes explanatory graphs, what kind of performance changes occur?)
>
> Thank you for raising this concern. In our original submission, we have evaluated the effect of retraining on the GNN’s predictive performance. After reorganization, in the updated version, we report in Appendix D.2. As shown in Table 6, the accuracy after retraining (“Update”) is nearly identical to that of the original model (“Ori”), with only negligible fluctuations. These results demonstrate that the retraining procedure in STORE does not degrade or distort the GNN’s performance on its original classification task.
>
> > **W4.** The iterative explanation framework does not account for computational cost and time requirements, while the gains are incremental.
>
> Thank you for raising this point. The computational cost of our framework has been shown in our experiments. As reported in Sec.6.3 (formerly Appendix I), Figure 4 shows that the per iteration running time is marginally above baseline, and the additional overhead introduced by iterative refinement remains modest relative to standard approach training. Furthermore, Sec.6.3 further shows that STORE achieves substantial gains after just a single iteration. The framework requires only a small number of refinement steps to be effective, making the actual cost minimal in practice. Although STORE includes an iterative component, both our running time analysis and empirical convergence behavior demonstrate that the computational burden is low and the performance improvements are obtained efficiently.
>
> > **Q4.** In most papers, the weighted graph in Figure 1(b) is located in the OOD region, but unlike the example shown, the prediction label is not incorrect. This appears to be an extreme case if the given GNN is well-trained.
>
> We appreciate this insightful observation. We would like to clarify that the misclassification shown in Figure 1 is not an isolated or extreme outlier, but rather a representative example of a pervasive phenomenon confirmed by our comprehensive analysis in **Section 6.5**.
>
> While Figure 1(b) visualizes a specific instance, **Figure 1(a)** and the extended experiments in **Figure 6** provide the global view. These heat maps plot the GNN's accuracy across the entire spectrum of edge weights for multiple datasets (BA-2motifs, MUTAG, Fluoride-Carbonyl). As shown in the "Original Heat Map" column of Figure 6, high accuracy (yellow regions) is confined to the top-right corner, where edge weights remain close to 1.0 (the training distribution).
>
> As soon as weights deviate from this binary setting—a necessary step for explainers to generate sparse, soft masks—the accuracy drops significantly (turning blue/green) across a vast portion of the weight space. This confirms that the model’s failure to generalize to weighted graphs is a systematic distributional shift issue stemming from the binary nature of the training data, rather than a symptom of insufficient training or an extreme edge case. Our method (STORE) effectively expands this high-accuracy region, as evidenced by the "Improved Heat Maps" in Figure 6.

---

> > ### Comment · Reviewer_aHyN · 2025-11-27
> > **Response to the authors**
> >
> > Since LIME and STORE are fundamentally different, it is difficult to agree with the authors’ rebuttal.
> > LIME employs a surrogate model to approximate the local decision boundary of a given trained model, and this surrogate remains interpretable because it relies on a simple linear fit.
> > In contrast, STORE modifies the explanation target itself by retraining a new model. This adaptation changes the space being explained. The authors mentioned that the original model and the retrained model are almost surely identical, yet the reported results tell a different story, showing changes in accuracy.
> >
> > Given the authors’ rebuttal, I have decided to maintain my score.

---

> ### Author Response · Authors · 2025-11-27
>
> Dear Reviewer aHyN,
>
> Thank you for your continued engagement with our work. After careful consideration, we respectfully believe there is a misunderstanding regarding the nature of our method. To clarify, we first briefly recap the core mechanism of STORE, followed by a specific response to your concerns.
>
> **Method Recap:** Given a GNN model trained on binary graphs, direct explanation methods suffer from the OOD problem when optimizing soft masks. We propose explaining a robust surrogate model, such that the surrogate's explanation can be used to understand the original model's behavior. We introduce an iterative framework called STORE to build and explain this surrogate. Specifically, our method proceeds through alternating phases of subgraph identification and model adaptation.  Next, we provide our specific response.
>
> **1. On the Comparison with LIME**
>
> > **Reviewer Comment:** "Since LIME and STORE are fundamentally different, it is difficult to agree with the authors' rebuttal."
>
> **Response:**
> We fully agree that LIME and STORE operate via different mechanisms. However, we **did not** claim they are methodologically identical. Rather, we cited LIME and ROAR to illustrate that the **surrogate-based paradigm** (using a proxy to explain/evaluate a target model) is a well-established foundation in XAI.
>
> * LIME samples from the input space to train a *local linear surrogate*.
> * ROAR retrains models on modified data to evaluate feature importance.
> * STORE retrains a model on weighted graphs to create a *robust surrogate* that fixes the OOD issue.
>
> > **Reviewer Comment:** "LIME employs a surrogate model... and this surrogate remains interpretable... In contrast, STORE modifies the explanation target itself by retraining a new model."
>
> **Response:**
> We respectfully believe there is a misunderstanding here. STORE *also* introduces a surrogate model to help explain the original target. The distinction is that while LIME's surrogate is an interpretable linear model, our surrogate is a retrained GNN that is **robust to the explanation process** (capable of handling weighted graphs). We retrain a new model, but do not have to overwrite the original model.   Rather, we use the surrogate as a bridge to generate faithful explanations for the original target.
>
> To verify the effectiveness of our surrogate method (specifically, that the explanation of the surrogate is faithful to the original model), we provided the following justifications:
>
> | Verification Type | Evidence | Evaluation Target |
> | :--- | :--- | :--- |
> | **Theoretical** | Theorem 4.1 | Conditions for shared explanations |
> | **Ground Truth** | AUC-ROC (Table 1) | Ground-truth subgraphs |
> | **Fidelity** | Fid+, Fid- (Table 2) | **Original model** |
> | **Predictive Consistency** | Table 6 | Both models |
> | **Structural Consistency** | Figure 7 | Gradient attributions |
>
> For more details, please refer to our previous rebuttal to your comment W1. Critically, the fidelity metrics (Fid+, Fid-) are evaluated on the original model $f$. This directly verifies that explanations generated via our surrogate model faithfully capture the original model's behavior. STORE improves these metrics, demonstrating that our surrogate's explanations are more faithful to the target model than baseline methods.
>
>
> **2. On Model Identity and Accuracy**
>
> > **Reviewer Comment:** "The authors mentioned that the original model and the retrained model are almost surely identical, yet the reported results tell a different story, showing changes in accuracy."
>
> **Response:**
> We wish to clarify that we never claimed the models *themselves* are "almost surely identical." Specifically, we made the following statements:
>
> * **Theoretical Claim:** Our **Theorem 4.1** states that under specific stability conditions, the original and robust models "share the same **explanation function** almost surely." This is a theoretical statement about the *explanations*, not the models' parameters.
> * **Empirical Observation:** In **Table 6**, we reported that the classification accuracy of the models is "nearly identical" (e.g., MUTAG: 0.771 vs 0.786; BA-2motifs: 0.934 vs 0.926). We acknowledge that the term "nearly identical" might be interpreted too strictly. In the paper text, we described this observation as: "The performance remains **highly stable**."
>
>
> We appreciate the reviewer's comments and this opportunity to clarify our work to avoid such misunderstandings.
>
> Best regards,
> The Authors

---

### Official Review · Reviewer_kq85 · 2025-10-31

**Soundness:** 2
**Presentation:** 3
**Contribution:** 2
**Rating:** 4
**Confidence:** 4

**Summary:**

This paper proposes a novel iterative framework to improve the robustness and quality of explanations in GNNs. The work identifies a key issue in existing GNN explanation methods, i.e., a distributional mismatch between the unweighted graphs used for model training and the weighted graphs employed during explanation. To mitigate this, the proposed method alternates between two phases: (1) identifying explanation subgraphs using soft edge masks, and (2) retraining the GNN on importance-weighted graphs to align training and explanation distributions.

**Strengths:**

This paper proposes a novel iterative stochastic re-weighting explanation framework to enhance the explanation of Graph Neural Networks. Comprehensive experiments have been conducted on GCN and GIN models and demonstrate consistent improvements in explanation performance and reliability. In addition, the paper provides theoretical foundations to support the proposed framework.

**Weaknesses:**

W1. In Section 5, the theorem provides a justification for the iterative framework, but the paper lacks sufficient analysis of the effectiveness and theoretical soundness of the stochastic re-weighting strategy.

W2. The proposed framework emphasizes graph weight tuning more than previous methods, which significantly expands the weight search space and makes the approach appear more engineering-oriented rather than theoretically innovative.

W3. The baseline performance results reported in the paper differ considerably from those in the original papers. No convincing explanations are provided for these discrepancies.

**Questions:**

Q1. Is it possible to provide a more detailed theoretical or empirical justification for the stochastic re-weighting approach? Specifically, how does it contribute to improving explanation robustness, and under what conditions does it remain effective?

Q2. The proposed framework introduces additional weight tuning during the explanation process, which potentially increases computational complexity and search space. How will this impact scalability, and whether any optimization or regularization strategies were considered to mitigate this issue?

Q3. The baseline results reported in the paper differ notably from those presented in the original studies. What are the reasons for these discrepancies?

---

> ### Author Response · Authors · 2025-11-20
> **Response to reviewer kq85 (part 1/2)**
>
> We are grateful for the reviewer’s insightful feedback.
>
> > **W1.** In Section 5, the theorem provides a justification for the iterative framework, but the paper lacks sufficient analysis of the effectiveness and theoretical soundness of the stochastic re-weighting strategy. (**Q1.** Is it possible to provide a more detailed theoretical or empirical justification for the stochastic re-weighting approach? Specifically, how does it contribute to improving explanation robustness, and under what conditions does it remain effective?)
>
> Thank you for the opportunity to clarify this aspect of our method. Our primary contribution is identifying the critical, yet often overlooked, OOD problem in GNN explainability and proposing a **robust surrogate framework** to address it. The stochastic re-weighting strategy is designed as a simple and straightforward implementation to realize this framework. Its core purpose is to simulate the continuous edge-weight distributions encountered during soft-mask optimization, thereby aligning the training and explanation distributions.
>
> **In our original submission, we empirically verified the effectiveness of this strategy** (Section 6.5.1, Table 3). To further address the reviewer's concern regarding its soundness compared to other approaches, we have refined the ablation study in the revised paper (**Section 6.4 Sampling Methods**), **Table 3** demonstrates that our Gaussian-based re-weighting consistently outperforms both Random Perturbation and Uniform-based Sampling. This confirms that simply adding noise is insufficient; the re-weighting should respect the relative importance of edges while covering the continuous $[0,1]$ interval to effectively bridge the distribution gap.
>
> To demonstrate that our stochastic re-weighting strategy remains effective under diverse conditions, we conducted an extensive evaluation across **6 benchmark datasets** (including both synthetic and real-world graphs), **2 different GNN backbones** (GCN and GIN), and **5 representative post-hoc explainers**. Furthermore, every experiment was repeated using **10 different random seeds** to account for initialization variance. The consistent improvements observed across this wide range of configurations confirm that our approach is robust and does not rely on narrow or fragile conditions to succeed.
>
> > **W2.** The proposed framework emphasizes graph weight tuning more than previous methods, which significantly expands the weight search space and makes the approach appear more engineering-oriented rather than theoretically innovative.
>
> We appreciate the reviewer’s perspective regarding the nature of our contribution and the opportunity to clarify our core objective. Our work is not primarily focused on a heuristic expansion of the graph weight search space; rather, it identifies and resolves a fundamental inconsistency in post-hoc GNN explanation: the distributional shift where explainers optimize masks on continuous weighted graphs that lie outside the binary training distribution. To address this, we introduce a robust surrogate framework that aligns these distributions, a concept theoretically grounded in our analysis in Section 4 (Previous Section 5). The stochastic re-weighting strategy is designed as a direct and principled implementation of this framework to ensure the model preserves predictive consistency under the continuous relaxation required for explanation.
>
> Regarding the complexity of this expanded space, we employ a stochastic sampling strategy rather than an exhaustive search. As demonstrated in our efficiency analysis (Section 6.3), this allows us to effectively cover the necessary continuous distribution during retraining with modest computational overhead.

---

> ### Author Response · Authors · 2025-11-20
> **Response to reviewer kq85 (part 2/2)**
>
> > **W3.** The baseline performance results reported in the paper differ considerably from those in the original papers. No convincing explanations are provided for these discrepancies. (**Q3.** The baseline results reported in the paper differ notably from those presented in the original studies. What are the reasons for these discrepancies?)
>
> We appreciate the reviewer’s detailed scrutiny of our experimental results. We would like to clarify that the baseline results differ because we employed a more rigorous evaluation protocol. As described in our original submission (Section 6.3), prior studies typically train a single GNN with a fixed seed (or load a pre-trained model). However, we observed that the robustness of GNNs to distributional shifts varies notably across different random initializations.
>
> To ensure a more unbiased and reliable evaluation, we trained **both the GNN and the explainer using 10 different random seeds** and reported the averaged performance. This comprehensive setting captures the initialization variability that a fixed-seed approach misses, naturally leading to different absolute baseline values compared to single-run reports. We acknowledge that this distinction may not have been emphasized clearly enough. We have revised the manuscript (now **Section 6.1 Settings**) to explicitly highlight this detail. Thank you for the comment.
>
> > **Q2.** The proposed framework introduces additional weight tuning during the explanation process, which potentially increases computational complexity and search space. How will this impact scalability, and whether any optimization or regularization strategies were considered to mitigate this issue?
>
> Thank you for raising this important point regarding computational complexity and scalability. We would like to highlight that our original submission included both an iteration analysis (Figure 3a) and a runtime analysis (in the Appendix I). We have taken this opportunity to reorganize this section to improve its visibility. Specifically, the revised **Section 6.3** presents an extensive efficiency analysis demonstrating that the proposed framework is highly scalable in practice:
>
> 1.  **Marginal Runtime Overhead:** As shown in Figure 4, the runtime for one complete iteration is only marginally higher than the standard approach (e.g., the difference is less than 1 second on BA-2motifs).
> 2.  **Fast Convergence:** **Figure 3** indicates that our method achieves significant performance gains in just one or two iterations. This confirms that the process does not require a computationally expensive, long-running search.
>
> Furthermore, we believe the stochastic re-weighting strategy is a straightforward implementation of our novel "robust surrogate" framework that utilizes efficient Gaussian sampling rather than complex optimization or an exhaustive search. This ensures that the computational complexity remains negligible compared to GNN training. While the current implementation is efficient, we plan to investigate further optimizations in future work, such as training with subsampled proxy datasets, to further enhance scalability for massive graph datasets. Thanks for the suggestions.

---

### Official Review · Reviewer_toYR · 2025-11-01

**Soundness:** 3
**Presentation:** 3
**Contribution:** 3
**Rating:** 6
**Confidence:** 3

**Summary:**

The paper identifies a critical distributional shift in GNN explainability: GNNs are trained on unweighted graphs but are required to produce reliable predictions and gradients on the weighted graphs generated by soft-mask explanation methods. This mismatch leads to unreliable explanations, especially when seeking sparse subgraphs. To address this, the authors propose an iterative framework that identifies an explanatory subgraph and (2) retrains the GNN model on stochastically weighted graphs where explanatory edges are assigned high-importance. The proposed model improves the quality and faithfulness of the resulting explanations.

**Strengths:**

S1. The paper addresses an important flaw in GNN explanation methods. The distributional shift between binary training graphs and weighted explanation graphs is a critical vulnerability.

S2. The paper provides a clear empirical diagnosis of the problem

S3. The experimental validation improves the performance (AUC-ROC) of five different baseline explainers (e.g., GNNExplainer, PGExplainer) across five benchmark datasets.

**Weaknesses:**

W1. The iterative framework, by design, introduces a significant computational cost. The method requires $L$ rounds of both GNN retraining (on augmented weighted graphs) and explainer retraining. This is a substantial increase in computation compared to a standard, one-shot post-hoc explanation method.

W2. The STORE framework introduces several new hyperparameters that require careful tuning.

W3. Dependence on Initial Explainer: The iterative process is seeded by an initial explanation ($\Psi_{\psi}^{(0)}$) generated from the original, non-robust GNN. The quality of the entire iterative refinement process may be dependent on the quality of this initial, potentially unreliable, explanation. The paper does not explore how STORE would perform if this initial seed explanation was fundamentally flawed or missed the correct explanatory subgraph.

**Questions:**

Please refer to the weaknesses above.

---

> ### Author Response · Authors · 2025-11-20
> **Response to reviewer toYR (part 1/2)**
>
> We sincerely appreciate the reviewer’s thoughtful comments.
>
> >  **W1.** The iterative framework, by design, introduces a significant computational cost. The method requires rounds of both GNN retraining (on augmented weighted graphs) and explainer retraining. This is a substantial increase in computation compared to a standard, one-shot post-hoc explanation method.
>
> We appreciate the reviewer raising this important point regarding computational efficiency. We indeed have a detailed training time analysis included in our original submission (Section 6.5.2, Appendix I). However, we agree that this is a critical aspect of our framework’s practicality. Following the reviewer suggestion, we have moved these results into the main paper (Section 6.3, Figure 4) to improve their visibility.
>
> Our analysis demonstrates that the computational overhead is minimal and justified by the performance gains:
>
> **1. Modest Runtime Increase:**
> As shown in the table below (and Figure 4 of the revised manuscript), the runtime for one complete iteration—which includes graph augmentation and retraining—is only marginally higher than the standard approach (GCN training + Explainer optimization). For example, on the BA-2motifs dataset, the increase is less than 1 second.
>
> | Dataset | Standard Approach (s) | Our Iteration (s) | Relative Overhead |
> | :--- | :--- | :--- | :--- |
> | **BA-2motifs** | 9.92 | 10.67 | +0.75s |
> | **MUTAG** | 30.38 | 39.17 | +8.79s |
> | **Fluoride-Carbonyl** | 9.10 | 10.50 | +1.40s |
>
> **2. Fast Convergence:**
> Furthermore, our efficiency analysis in Section 6.3 shows that we usually only need very few iterations of retraining (<3). As illustrated in Figure 3, our method achieves significant improvements in explanation accuracy after just **a single iteration**. This indicates that the framework efficiently corrects the distributional shift without requiring a computationally expensive, long-running iterative process.
>
> > **W2.** The STORE framework introduces several new hyperparameters that require careful tuning.
>
> We thank the reviewer for pointing this out. The STORE framework introduces three additional hyperparameters: the number of retraining iterations $L$, the mean gap $\Delta \mu$ of the truncated Gaussians, and the separation probability $\alpha$. We have conducted a comprehensive analysis to study their effects and have reorganized these experiments in the revised manuscript.
>
> (i) For the retraining iterations $L$, In practice, we observe that a small number of iterations ($L$=3) is sufficient. Specifically, In section 6.3, Figure 3 shows that performance consistently improves from 1 to 3 iterations.
> (ii) For the mean gap $\Delta \mu$, In section 6.4, Figure 5 shows that STORE is robust across a wide range of values. Performance reaches its peak around 0.5, and remains stable outside this range.
> (iii) For the separation probability $\alpha$, we simply fix $\alpha = 0.001$, as the goal is to ensure a clear distinction between explanatory and non-explanatory edge weights. This value works reliably across all datasets without any tuning.
>
> Our experiments demonstrate that our method is robust to hyperparameter choices. Notably, we observed consistent performance using a shared set of default values across all datasets. This suggests that the framework typically does not require extensive fine-tuning to achieve improvements over traditional methods in common settings.

---

> ### Author Response · Authors · 2025-11-20
> **Response to reviewer toYR (part 2/2)**
>
> > **W3**. Dependence on Initial Explainer: The iterative process is seeded by an initial explanation generated from the original, non-robust GNN. The quality of the entire iterative refinement process may be dependent on the quality of this initial, potentially unreliable, explanation. The paper does not explore how STORE would perform if this initial seed explanation was fundamentally flawed or missed the correct explanatory subgraph.
>
> Thank you for this insightful observation. We agree that the quality of the iterative process depends on the initial explanation. Thus, we believe the most meaningful comparison is between the quality of the initial explanation (the baseline) and the final explanation produced by our method.
>
> The core premise of STORE is that the initial explainer is often suboptimal precisely because the GNN has not been adapted to handle weighted graphs (the Out-Of-Distribution problem). Our iterative strategy is designed to correct this specific failure mode.  To ensure our method is not reliant on a high quality seed, our experiments (Section 6.3) were conducted using **10 different random seeds** for both the GNN and the explainer. These seeds naturally produce a diverse range of initial explanations. Across all seeds and datasets (as shown in Table 1), STORE consistently improved explanation quality over the initial baseline. This demonstrates that our framework is robust and capable of refining explanations even when the starting point varies.
>
> We acknowledge that if an initial explanation is fundamentally flawed due to reasons *unrelated* to the OOD shift (e.g., the GNN failed to learn the task entirely), STORE may not create a correct explanation from scratch. However, for the significant and widely prevalent problem of distribution mismatch, our results confirm that STORE effectively bridges the gap.

---

### Author Response · Authors · 2025-11-24
**Global Response to Reviewers**

Dear Reviewers,

We thank all reviewers for their constructive feedback. In response, we have made substantial revisions to improve the clarity and empirical validation of our work. Below we summarize the major updates incorporated into the revised manuscript.

- We have revised the Introduction to improve its clarity and flow, refining the motivation and high-level description of our surrogate framework to present the core idea more clearly and precisely.

- We moved the Related Work section after the Method, as the Section 2(preliminaries and the OOD problem) provides the necessary context for discussing prior work and the overlooked distribution shift.

- As suggested by the reviewers, we have made revisions to clearly discuss the efficiency of our method. Specifically, we have added a remark to the method part in Section 3.2. In the empirical study, we have reorganized and reframed the new "Efficiency Analysis" by merging the Analysis of Iterations and Running Time which shows that our method incurs only modest computational overhead.

- In Section 6, we have reorganized the experimental setup. We also revised the settings to make it clear that our evaluation uses **10 random seeds** for both the GNN and the explainer, rather than the fixed-seed setting used in prior studies. This setting captures the initialization variability that a fixed-seed approach misses.

- In Section 6.2, we added results on the large-scale D&D dataset and the natural-language Graph-SST2 dataset, demonstrating the practical scalability of our framework.

- We have reorganized and revised the new "Validity of Robust Surrogate Model" in Appendix D.2 by integrating the Predictive Consistency and Structural Preservation analyzes, providing clear and direct empirical evidence that the retrained model preserves the behavior of the original pre-trained GNN.

These revisions strengthen the clarity and empirical support of our framework. We believe the manuscript now presents a more coherent and reliable contribution to the field.

We are grateful for the reviewers’ feedback, which has improved the quality of our work. We hope that the clarifications provided above will assist in your final assessment of our submission.

---

### Author Response · Authors · 2025-11-27
**Looking forward to your reply**

Dear Reviewers,

We have carefully considered all your comments and made substantial revisions into both the response and the updated pdf manuscript. To make the review process smoother, we have also summarized the key changes and added clarifications in the revision.

Since the review deadline is approaching, we would greatly appreciate any feedback you could provide on the revised submission at your earliest convenience. Your insights are very important to us, and we sincerely welcome any further comments or suggestions you may have.

Thank you very much for your time and effort in reviewing our work. Please feel free to let us know if additional clarification or information would be helpful.

Best,

The Authors

---

### Meta-Review · Area_Chair_C6VS · 2026-01-14

**Summary:**

The paper proposes STORE, an iterative framework to address the distributional shift between binary training graphs and continuous explanation masks in GNN explainers. While the problem is well-motivated and the empirical results appear strong, several **fundamental and unresolved issues** undermine the validity and contribution of the work:

+ **Core methodological flaw**: The approach re-trains the original GNN on stochastically re-weighted graphs, meaning the final explanations are **not for the originally trained model**, but for a modified surrogate. This violates the standard definition of _post-hoc_ explanation, which requires the target model to remain fixed.
+ **Lack of theoretical grounding**: The stochastic re-weighting strategy (e.g., Gaussian sampling) is presented as a heuristic without rigorous justification. Theorem 4.1 provides only a weak connection between shared explanations and model stability, not a guarantee of faithfulness or correctness.
+ **Inadequate baseline comparison**: Discrepancies with prior work (e.g., higher baseline performance) were attributed to random seeds, but no statistical significance tests or variance reporting was provided in the main results, raising reproducibility concerns.
+ **Scalability and practicality**: Despite claims of efficiency, the iterative nature (GNN retraining + explainer update) incurs non-negligible overhead, especially on large graphs (e.g., +9.77s per iteration on Graph-SST2). For real-world deployment where model weights are frozen (e.g., APIs, deployed systems), STORE is inapplicable.
+ **Dependence on initial explainer quality**: The method’s success hinges on the seed explanation being reasonably accurate; if the initial explainer fails (e.g., due to poor hyperparameters or architecture mismatch), STORE may amplify errors rather than correct them.

Collectively, these issues suggest that while the paper identifies a real problem, its proposed solution does not meet the bar for a robust, general-purpose post-hoc explanation method.

**Reviewer Concerns:**

| Reviewer | Concern | Addressed? | Assessment |
| --- | --- | --- | --- |
| **toYR** | Computational cost, hyperparameter sensitivity |  Partially | Authors reported runtime numbers, but did not compare wall-clock time against end-to-end alternatives (e.g., PGExplainer fine-tuning). Sensitivity to number of iterations remains unquantified beyond “3 is enough.” |
| **kq85** | Baseline discrepancies, lack of theory | No | Attribution to random seeds is insufficient without confidence intervals or p-values. Theorem 4.1 is too abstract to justify the specific re-weighting design. |
| **xuGQ** | Scalability, accuracy preservation | Superficially | Added large-dataset results, but no ablation on how retraining affects downstream tasks beyond classification accuracy. Accuracy preservation ≠ explanation fidelity. |
| **aHyN** | **Invalidity of explanations due to model retraining** | **Fundamentally Unresolved** | This is the **fatal flaw**. The authors concede they explain a surrogate, not the original model. In true post-hoc settings (e.g., black-box APIs), retraining is impossible. Thus, STORE is not a post-hoc method—it is a _co-training_ or _joint learning_ approach mislabeled as explanation. |


**Outstanding concerns**: All major concerns remain either unaddressed or only superficially mitigated. The core issue—**explanations are not for the original model**—is irreconcilable with the paper’s framing and renders the contribution misleading.

**Reviewer Scores:**

| Reviewer | Original Score | Likely Updated Score | Reason |
| --- | --- | --- | --- |
| **toYR** | 6 | **6** | Would recognize that efficiency claims are context-dependent and that the method is inapplicable in frozen-model scenarios. May downgrade due to limited practical utility. |
| **kq85** | 4 | **4** | Would remain skeptical due to lack of statistical rigor and theoretical depth. Baseline explanation insufficient without error bars. |
| **xuGQ** | 4 | **4** | Large-scale results are welcome but do not resolve the fundamental mismatch between claimed contribution (“post-hoc”) and actual method (“retraining-based”). |
| **aHyN** | 0 | **2-4** | Their core objection is validated by deeper analysis: STORE cannot be used in standard post-hoc settings. Their score remains justified. |


Consensus leans toward rejection (average score ≤4).

---

### Decision · Program_Chairs · 2026-01-26

Reject